DOI: 10.1038/s41467-018-06446-0　　**OPEN**

# Bone protection by inhibition of microRNA-182

Kazuki Inoue[1,2], Zhonghao Deng[1], Yufan Chen[3], Eugenia Giannopoulou[1,4], Ren Xu [5], Shiaoching Gong[6], Matthew B. Greenblatt[5], Lingegowda S. Mangala[7,8], Gabriel Lopez-Berestein[8,9], David G. Kirsch[10], Anil K. Sood[7,8,11], Liang Zhao[3] & Baohong Zhao [1,2,12]

Targeting microRNAs recently shows significant therapeutic promise; however, such progress is underdeveloped in treatment of skeletal diseases with osteolysis, such as osteoporosis and rheumatoid arthritis (RA). Here, we identified miR-182 as a key osteoclastogenic regulator in bone homeostasis and diseases. Myeloid-specific deletion of miR-182 protects mice against excessive osteoclastogenesis and bone resorption in disease models of ovariectomy-induced osteoporosis and inflammatory arthritis. Pharmacological treatment of these diseases with miR-182 inhibitors completely suppresses pathologic bone erosion. Mechanistically, we identify protein kinase double-stranded RNA-dependent (PKR) as a new and essential miR-182 target that is a novel inhibitor of osteoclastogenesis via regulation of the endogenous interferon (IFN)-β-mediated autocrine feedback loop. The expression levels of miR-182, PKR, and IFN-β are altered in RA and are significantly correlated with the osteoclastogenic capacity of RA monocytes. Our findings reveal a previously unrecognized regulatory network mediated by miR-182-PKR-IFN-β axis in osteoclastogenesis, and highlight the therapeutic implications of miR-182 inhibition in osteoprotection.

---

[1] Arthritis and Tissue Degeneration Program, The David Z. Rosensweig Genomics Research Center, Hospital for Special Surgery, New York, 10021 NY, USA. [2] Department of Medicine, Weill Cornell Medical College, New York, 10065 NY, USA. [3] Department of Orthopedic Surgery, Nanfang Hospital, Southern Medical University, Guangzhou 510515, China. [4] Biological Sciences Department, New York City College of Technology, City University of New York, Brooklyn, 11201 NY, USA. [5] Department of Pathology and Laboratory Medicine, Weill Cornell Medical College, New York, 10065 NY, USA. [6] Department of Molecular Biology, The Rockefeller University, New York, 10065 NY, USA. [7] Department of Gynecologic Oncology and Reproductive Medicine, The University of Texas MD Anderson Cancer Center, Houston, 77030 TX, USA. [8] Center for RNA Interference and Noncoding RNA, The University of Texas MD Anderson Cancer Center, Houston, 77030 TX, USA. [9] Department of Experimental Therapeutics, The University of Texas MD Anderson Cancer Center, Houston, 77030 TX, USA. [10] Department of Radiation Oncology and Department of Pharmacology and Cancer Biology, Duke University Medical Center, Durham, 27710 NC, USA. [11] Department of Cancer Biology, The University of Texas MD Anderson Cancer Center, Houston, 77030 TX, USA. [12] Graduate Program in Biochemistry Cell & Molecular Biology, Weill Cornell Graduate School of Medical Sciences, New York, 10065 NY, USA. Correspondence and requests for materials should be addressed to B.Z. (email: zhaob@hss.edu)

Bone destruction is a major characteristic and severe consequence of multiple skeletal diseases, including osteoporosis and inflammatory arthritis, and significantly reduces the quality of life of these patients and increases their risk of disability. Osteoclasts are the sole specialized bone-resorbing cells, which play an indispensable role as an exclusive pathogenic factor in bone destruction associated diseases, such as rheumatoid arthritis (RA) and postmenopausal osteoporosis[1–6]. MicroRNAs (miRNAs) function as essential regulators of a variety of biological and pathological settings, and have recently gained increasing clinical attention as promising therapeutic targets or biomarkers[7–14]. Functional studies of miRNAs have shown promise in preclinical development and several clinical trials in cancer, metabolic and infectious diseases, highlighting miRNA-based therapeutics toward a new era for disease treatment[15–17]. Nonetheless, such progress in treatment of skeletal diseases has not been made. Although the investigation of the role of miRNAs in bone biology is expanding[18–21], key miRNAs regulating osteoclastogenesis and their functions in bone diseases still remain underexplored.

miRNAs are evolutionarily conserved small noncoding RNAs consisting of ~22 nucleotides that are spliced from longer precursor transcripts. miRNAs target specific mRNAs via imperfect complementary binding but with a perfect base pairing between the miRNA "seed region" (nucleotides 2–7 of the miRNA), and the targeted sequences of mRNAs. miRNAs repress gene expression at the posttranscriptional level by promoting degradation and/or inhibiting translation of specific target mRNAs. The expression and function of miRNAs are highly specific to cell and tissue types and correlated with various stresses or disease settings[7,9–11,22,23]. The functional diversity of each miRNA appears to attribute to its targets and corresponding molecular networks in different scenarios. Our initial exploration of miRNAs involved in osteoclastogenesis showed that miR-182 is a new regulator in tumor necrosis factor (TNF)-induced inflammatory osteoclast differentiation in vitro[24]. The important function of miR-182 in cell growth, cell fate decision, cancer, and T lymphocyte expansion was just recently appreciated[25–34]. However, the role of miR-182 in vivo in physiological bone metabolism and pathological conditions such as that occur in osteoporosis and inflammatory arthritis has not yet been elucidated.

In this study, we identify miR-182 as a key positive regulator of osteoclastogenesis in both physiological and disease conditions and provide new mechanisms for the regulation of osteoclastogenesis driven by the miR-182-PKR-IFN-β axis. Targeting miR-182 and its downstream targets may represent an attractive alternative or complementary therapeutic approach to prevent pathologic bone destruction.

## Results

**miR-182 regulates osteoclastogenesis and bone homeostasis**. We first examined the expression levels of miR-182 during osteoclastogenesis using bone marrow-derived macrophages (BMMs) as primary osteoclast precursors. In response to the master osteoclastogenic inducer RANKL, miR-182 expression was increased during a time course of osteoclast differentiation (Fig. 1a). We next applied complementary genetic approaches and generated osteoclastic miR-182 knockout and transgenic mice. We deleted *Mir182* specifically in myeloid lineage osteoclast precursors by crossing *Mir182^{flox/flox}* mice[30] with *LysMcre* mice, which express Cre under the control of the myeloid-specific lysozyme M promoter, to generate *Mir182^{flox/flox}LysMcre*(+) (hereafter referred to as *Mir182^{ΔM/ΔM}*). Their littermates with a *Mir182^{+/+}LysMcre*(+) genotype (hereafter referred to as wild type (WT)) are used as the controls. As a complementary genetic

approach, we generated myeloid lineage osteoclast precursor conditional *Mir182* transgenic (*Mir182^{mTg}*) mice by crossing *LysMcre* mice with LoxP-STOP-LoxP (LSL)–miR-182 mice[30]. The littermate *LysMcre* mice were used as the Control. Compared to the WT cell cultures, miR-182 deficiency in the *Mir182^{ΔM/ΔM}* BMM cultures strikingly inhibited TRAP-positive multinucleated osteoclast formation induced by RANKL (Fig. 1b). Inhibition of osteoclastogenesis by miR-182 deficiency was also observed in the presence of RANKL with different concentrations (Supplementary Fig. 1). In parallel with suppressed generation of TRAP-positive polykaryons, the expression of the key osteoclastogenic transcription factors *Nfatc1* (encoding NFATc1) and *Prdm1* (encoding Blimp1) as well as osteoclast marker genes *Acp5* (encoding TRAP), *Ctsk* (encoding cathepsin K) and *Itgb3* (encoding β3 integrin) was dramatically decreased in RANKL-treated *Mir182^{ΔM/ΔM}* cells relative to the WT control cells (Fig. 1c). Conversely, overexpression of miR-182 in BMMs derived from *Mir182^{mTg}* mice significantly augmented the osteoclastogenic program indicated by TRAP-positive osteoclast differentiation and corresponding osteoclastic gene expression compared with the control cell cultures (Supplementary Fig. 2a–d). Human miR-182 was also induced by RANKL during osteoclastogenesis using CD14-positive peripheral blood mononuclear cells (PBMCs)-derived human osteoclast precursors (Fig. 1d). Downregulation of miR-182 by a specific inhibitor in human osteoclast precursors resulted in significantly decreased osteoclastogenesis (Fig. 1e) and the osteoclastic gene expressions of NFATC1, PRDM1, CALCR (encoding calcitonin receptor), and ITGB3 (Fig. 1f). These results indicate that miR-182 is an important positive regulator promoting osteoclastogenesis. The osteoclastogenic function of miR-182 is conserved in human cells.

We next analyzed the bone phenotype of the osteoclastic miR-182 knockout and transgenic mice. Microcomputed tomographic (μCT) analyses showed that *Mir182^{ΔM/ΔM}* mice exhibited a strong osteopetrotic phenotype indicated by markedly increased bone mass (Fig. 1g, h), trabecular bone volume, bone mineral density (BMD), connectivity density (Conn-Dens.), trabecular bone number, and decreased trabecular bone spacing compared to WT littermate control mice (Fig. 1h). The cortical bone thickness was not altered by osteoclastic miR-182 deficiency (Supplementary Fig. 3a). In contrast, *Mir182^{mTg}* mice showed an osteoporotic bone phenotype accompanied by decreased bone mass (Supplementary Fig. 4a), trabecular bone volume, bone mineral density, connectivity density, trabecular bone number, and increased trabecular bone spacing compared to their corresponding littermate controls (Supplementary Fig. 4b). Furthermore, miR-182 affects osteoclast formation in vivo evidenced by lower osteoclast numbers and smaller osteoclast surface areas observed in the *Mir182^{ΔM/ΔM}* mice than in the WT controls (Fig. 1i, j). On the other hand, osteoclastic deficiency of miR-182 does not affect bone formation rate, osteoblast numbers and osteoblast surfaces (Supplementary Fig. 3b, c). Taken together of the complementary bone phenotypes of osteoclastic miR-182 knockout and transgenic mice as well as the impact of miR-182 on osteoclastogenesis, these data demonstrate that miR-182 is a key osteoclastogenic regulator and plays an essential role in the maintenance of bone homeostasis.

**miR-182 plays a crucial role in pathological bone loss**. We next turned to investigate the significance of miR-182 in pathological bone loss. With the consideration that the common bone diseases associated with bone loss are osteoporosis and inflammatory arthritis, we developed the ovariectomy (OVX) model in *Mir182^{ΔM/ΔM}* mice to study the role of miR-182 in estrogen deficiency induced osteoporosis, which mimics postmenopausal

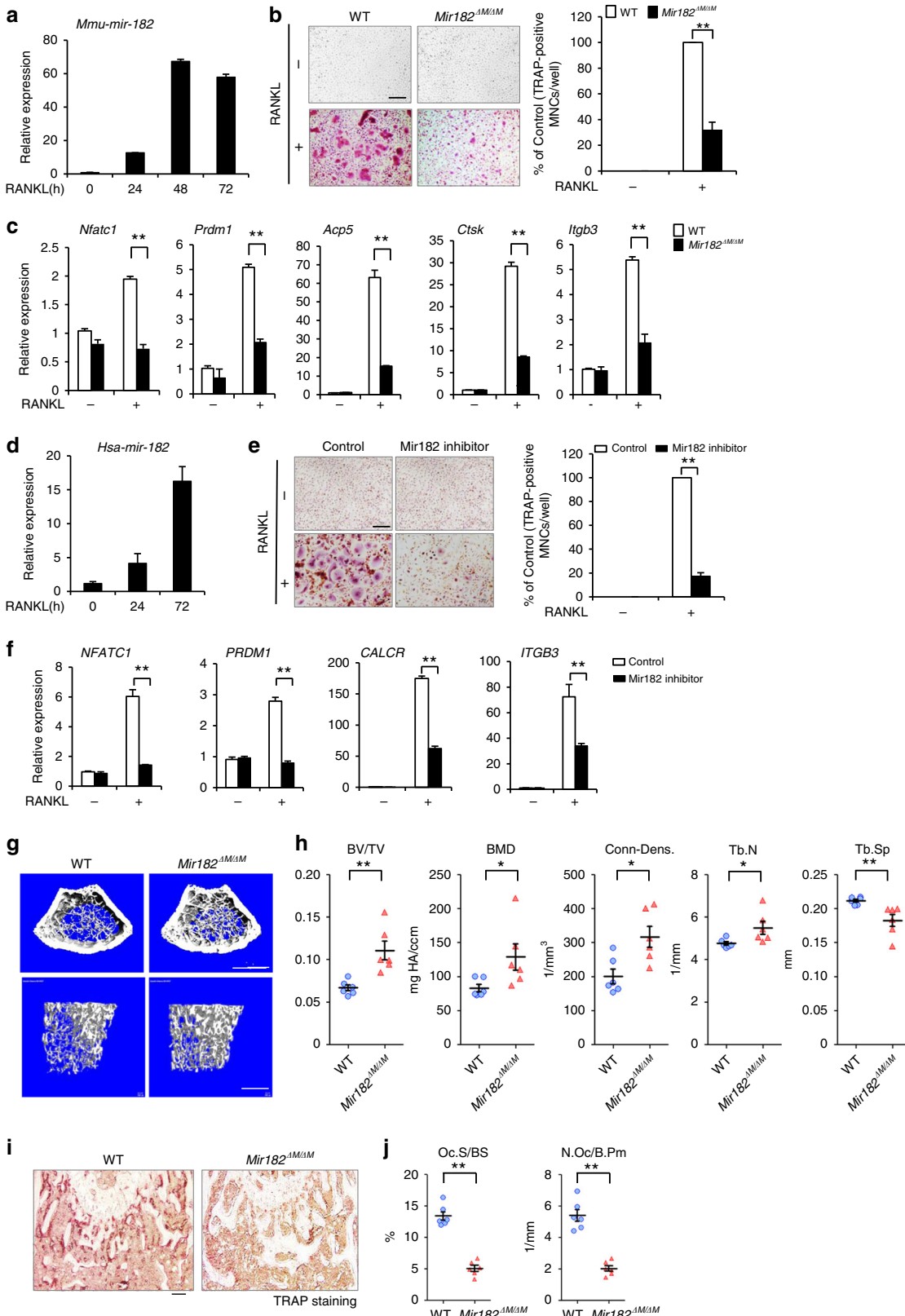

bone loss, and an inflammatory arthritis model to study the role of miR-182 in inflammatory bone resorption.

In the OVX-induced osteoporosis model, in order to assess the success of the OVX, uterine weight was measured 5 weeks after surgery. The uterine weight was decreased approximately 75% in OVX groups compared with the sham groups (Fig. 2a), indicating

effective estrogen depletion. As expected in the WT mice, OVX significantly reduced bone mass indicated by almost 50% decrease of BV/TV, BMD and Conn-Dens. and 25% decrease of trabecular number, and about 30% increase of trabecular spacing compared with the sham group (Fig. 2b, c columns 1 vs. 2) revealed by the μCT analysis. Strikingly, the bone mass in $Mir182^{\Delta M/\Delta M}$ mice was

**Fig. 1** miR-182 is a key positive regulator of osteoclastogenesis and bone homeostasis. **a** Mature mouse miR-182 (*mmu-mir-182*) expression during RANKL-induced osteoclastogenesis. **b** Osteoclast differentiation using BMMs derived from WT and *Mir182*$^{\Delta M/\Delta M}$ mice stimulated with RANKL for three days. TRAP staining (left panel) was performed and the area of TRAP-positive MNCs (≥3 nuclei/cell) per well relative to the WT control was calculated (right panel). TRAP-positive cells appear red in the photographs. **c** Quantitative real-time PCR analysis of mRNA expression of *Nfatc1* (encoding NFATc1), *Prdm1* (encoding Blimp1), *Acp5* (encoding TRAP), *Ctsk* (encoding cathepsin K), and *Itgb3* (encoding β3 integrin) during osteoclastogenesis using BMMs from the WT and *Mir182*$^{\Delta M/\Delta M}$ mice treated with or without RANKL for 5 days. **d** Mature human miR-182 (*hsa-mir-182*) expression during RANKL-induced human osteoclastogenesis using human CD14(+) PBMCs as osteoclast precursors. **e** Human osteoclast differentiation induced by RANKL for three days using human CD14(+) monocytes that were transfected with the control or miR-182 Inhibitor. TRAP staining (left panel) was performed and the area of TRAP-positive MNCs (≥3 nuclei/cell) per well relative to the Control was calculated (right panel). TRAP-positive cells appear red in the photographs. **f** Quantitative real-time PCR analysis of mRNA expression of *NFATc1*, *PRDM1*, *CALCR* (encoding calcitonin receptor), and *ITGB3* from 3 day-RANKL-stimulated human CD14(+) monocytes transfected with the control or miR-182 inhibitor. **g** μCT images and **h** bone morphometric analysis of trabecular bone of the distal femurs isolated from 10-week-old littermate male WT and *Mir182*$^{\Delta M/\Delta M}$ mice (*n* = 6/group). **i** TRAP staining and **j** histomorphometric analysis of histological sections obtained from the metaphysis region of distal femurs of 10-week-old male WT and *Mir182*$^{\Delta M/\Delta M}$ mice. BV/TV, bone volume per tissue volume; BMD, bone mineral density; Conn-Dens., connectivity density; Tb.N, trabecular number; Tb.Sp, trabecular separation; Oc.S/BS, osteoclast surface per bone surface; N.Oc/B.Pm, number of osteoclasts per bone perimeter. **b, c, e, f, h, j** $^*p < 0.05$; $^{**}p < 0.01$ by Student's *t* test. Error bars: **a–f** Data are mean ± SD. **h, j** Data are mean ± SEM. Scale bars: **b, e** 200 μm; **g** 500 μm; **i** 100 μm

not decreased by OVX (Fig. 2b, c columns 3 vs. 4), indicating that miR-182 deficiency completely protected *Mir182*$^{\Delta M/\Delta M}$ mice against OVX-induced bone loss (Fig. 2b, c columns 3 vs. 4). Notably, this protection is not attributed to the higher basal bone volume in *Mir182*$^{\Delta M/\Delta M}$ mice than the WT mice, but because miR-182 deficiency counteracts the bone loss induced by OVX in *Mir182*$^{\Delta M/\Delta M}$ mice (Fig. 2b, c columns 3 vs. 4). Bone histomorphometric analysis further showed significantly larger osteoclast numbers and surfaces in the WT OVX group than the sham group (Fig. 2d, e columns 1 vs. 2). In contrast, OVX failed to stimulate pathologic osteoclastogenesis in *Mir182*$^{\Delta M/\Delta M}$ mice (Fig. 2d columns 3 vs. 4) and the osteoclastic parameters, including osteoclast numbers and surfaces, remained comparable between the sham and OVX of *Mir182*$^{\Delta M/\Delta M}$ mice (Fig. 2e columns 3 vs. 4). Osteoclastic miR-182 deficiency did not affect cortical thickness, bone formation rate, osteoblast numbers, and osteoblast surfaces in the sham or OVX mice (Supplementary Fig. 5a–c). These results indicate that osteoclastic deletion of miR-182 appears to enable osteoclast lineage cells to become "tolerant" to environmental osteoclastogenic stimuli, such as OVX, thereby preventing pathologic bone loss.

In inflammatory settings, osteoclast precursors are generally influenced by both RANKL and inflammatory cytokines, such as TNF-α, which promotes osteoclast formation by priming osteoclast precursors or acting in synergy with RANKL[35–41]. We found that miR-182 deficiency dramatically repressed osteoclast formation in the TNF-α or RANKL priming conditions (Supplementary Fig. 6), suggesting a positive regulatory role of miR-182 in osteoclastogenesis and bone resorption under inflammatory conditions. To further test this hypothesis in vivo, we first used a well-established inflammatory calvarial osteolysis mouse model induced by TNF-α or lipopolysaccharides (LPS)[42–46]. Phosphate buffered saline (PBS) injection as a negative control did not induce resorptive pit formation on the calvarial bone surfaces (data not shown). Administration of TNF-α to the *Mir182*$^{mTg}$ calvarial periosteum resulted in significantly enhanced erosions, identifiable by μCT analysis of the resorptive pits formed on the calvarial bone surface (Fig. 2f, g), and markedly increased osteoclast formation shown in the histological slices of the calvarial bones compared to the control mice (Fig. 2h, i). In the LPS-induced inflammatory bone resorption model, deletion of miR-182 in the *Mir182*$^{\Delta M/\Delta M}$ mice almost completely inhibited LPS-induced extensive bone destruction and osteoclast formation as observed in the WT calvarial bones (Fig. 2j–m). Next, we tested the role of miR-182 in a more pathologically relevant model that mimics human diseases with characteristics of

inflammatory bone destruction. To do this, we chose K/BxN serum-induced arthritis model[47], which has been widely used to study inflammatory peri-articular bone erosion induced by the inflammatory cytokines in the serum. This model can bypass the need to induce autoimmunity and thus allows investigation of inflammatory bone resorption during the inflammatory effector phase of arthritis. Deficiency of miR-182 in the *Mir182*$^{\Delta M/\Delta M}$ mice resulted in striking suppression of peri-articular bone erosion, osteoclast numbers and surfaces in resorption sites (Fig. 2n, o) of the tarsal joints in the inflammatory arthritis model when compared with the WT control mice. The clinical course of inflammatory arthritis indicated by the joint swelling was not diminished during the 10 day course except for bone erosion in *Mir182*$^{\Delta M/\Delta M}$ mice (Supplementary Fig. 7), indicating that miR-182 does not significantly impact inflammation in this model but prominently affects osteoclast formation and bone erosion.

**Inhibition of miR-182 has therapeutic significance.** Our genetic evidence inspired us to test the efficacy of therapeutic targeting of miR-182 to prevent bone loss for translational implications. To provide a proof of concept experiment to test this hypothesis, we chose nanoparticles as an effective and feasible in vivo delivery method for an miR-182 inhibitor[24], which has been validated to specifically target miR-182. Chitosan (CH) has been extensively used as drug delivery vehicles for small nucleotide oligos in preclinical studies[15,20,48–50]. Prompted by the recent work using CH-nanoparticles to effectively deliver small miRNAs in vivo to successfully target osteoclasts[20], we used the same optimized packaging formula, in which the small RNA oligos have the highest bio-distribution in bone marrow and target osteoclast lineage efficiently[20]. In vivo injection of CH-nanoparticles did not alter mouse body weight and bone mass (Supplementary Fig. 8a–d), indicating the absence of obvious toxicity or effects on bone from CH-nanoparticles per se.

In the OVX-induced osteoporosis model developed in the C57BL/6 mice, successful OVX was first validated by the uterine atrophy in all OVX mice (Fig. 3a). Severe osteoporosis was developed in the OVX mice injected with the CH-nanoparticles containing the corresponding control RNA oligos 5 weeks after surgery (Fig. 3b, c, columns 1 vs. 3). Strikingly, μCT analysis revealed that treatment with CH-nanoparticles containing miR-182 inhibitor oligos prevented the OVX-induced bone loss (Fig. 3b, columns 3 vs. 4; 2 vs. 4), assessed by the bone mass parameters BV/TV, BMD, Tb.N, Tb.Th, and Tb.Sp, to a similar level relative to the sham groups (Fig. 3c, columns 3 vs. 4; 2 vs. 4). In the sham groups, CH-nanoparticles containing miR-182

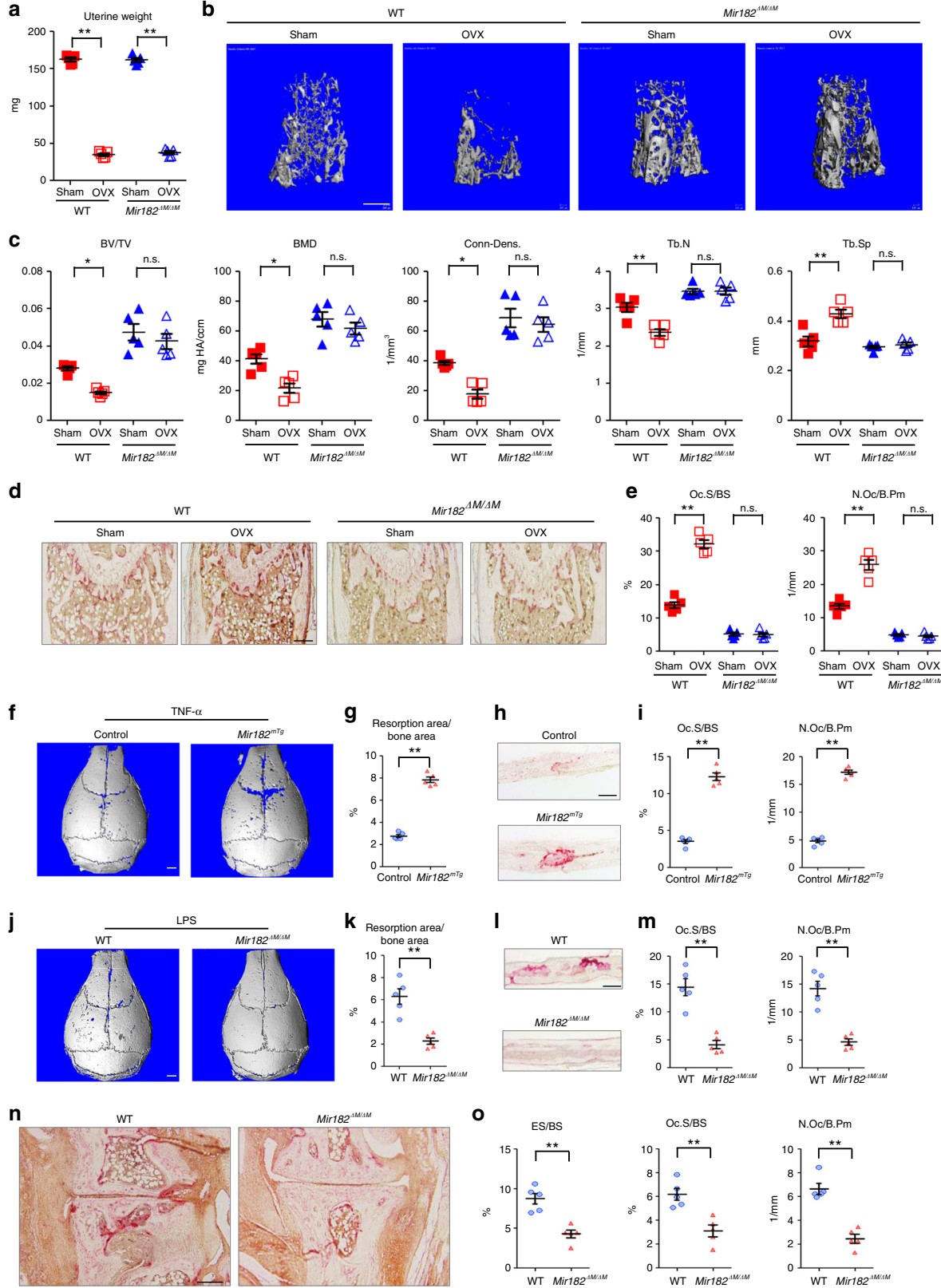

inhibitors did not affect basal bone mass during the treatment period (Fig. 3b, c, columns 1 vs. 2). Moreover, CH-nanoparticles containing miR-182 inhibitors drastically suppressed OVX-induced excessive osteoclast formation (Fig. 3d, e). The osteoclast numbers and surfaces in the mice treated with CH-nanoparticles containing miR-182 inhibitors were similar between the sham

and OVX groups (Fig. 3d, e). In addition, CH-nanoparticles containing miR-182 inhibitors did not affect cortical thickness, bone formation rate, osteoblast numbers and osteoblast surfaces in the OVX model (Supplementary Fig. 9a–c). Thus, inhibition of miR-182 implies an effective strategy in preventing bone loss induced by estrogen deficiency.

**Fig. 2** miR-182 deficiency protects mice from pathologic bone loss induced by OVX or inflammatory arthritis. **a**–**e** 10-week-old female WT and $Mir182^{\Delta M/\Delta M}$ mice were subjected to OVX or sham surgery and analyzed 5 weeks after surgery. **a** Uterine weight, **b** μCT images, and **c** bone morphometric analysis of trabecular bone of the distal femurs isolated from the WT and $Mir182^{\Delta M/\Delta M}$ mice with sham or OVX surgery ($n = 5$/group). **d** TRAP staining and **e** histomorphometric analysis of histological sections obtained from the metaphysis region of distal femurs isolated from the indicated mice. BV/TV, bone volume per tissue volume; BMD, bone mineral density; Conn-Dens., connectivity density; Tb.N, trabecular number; Tb.Sp, trabecular separation; Oc.S/BS, osteoclast surface per bone surface; N.Oc/B.Pm, number of osteoclasts per bone perimeter. **f**–**i** TNF-induced supracalvarial osteolysis model. **f** μCT images of the surface of whole calvaria. **g** quantification of the resorption area of the calvaria. **h** TRAP staining of calvarial histological sections and **i** histomorphometric analysis of calvarial slices obtained from the Control and $Mir182^{mTg}$ mice after the application of TNF-α daily for five days to the calvarial periosteum. $n = 5$ per group. **j**–**m** LPS-induced supracalvarial osteolysis model. **j** μCT images of the surface of whole calvaria, **k** quantification of the resorption area of the calvaria, **l** TRAP staining of calvarial histological sections, and **m** histomorphometric analysis of calvarial slices obtained from the WT and $Mir182^{\Delta M/\Delta M}$ mice five days after the application of LPS to the calvarial periosteum. $n = 5$ per group. **n** TRAP staining of histological sections of tarsal joints and **o** histomorphometric analysis of the tarsal joint sections obtained from the indicated mice that developed K/BxN serum-induced arthritis. ES/BS, erosion surface per bone surface; Oc.S/BS, osteoclast surface per bone surface; N.Oc/B.Pm, number of osteoclasts per bone perimeter. **a**, **c**, **e** $^*p < 0.05$; $^{**}p < 0.01$; n.s. not statistically significant by two-way ANOVA. **g**, **i**, **k**, **m**, **o** $^{**}p < 0.01$ by Student's $t$ test. Error bars: data are mean ± SEM. Scale bars: **b** 500 μm; **d, h, l, n** 200 μm; **f, j** 1.0 mm

Compared with the CH-nanoparticles containing the control RNA oligos, treatment of the mice with the CH-nanoparticles containing miR-182 inhibitor oligos starting from the early stage of inflammatory arthritis drastically suppressed osteoclast formation indicated by nearly 70% decrease in osteoclast numbers and 64% decrease in osteoclast surface areas, and as a consequence effectively alleviated approximately 67% of inflammatory arthritic bone resorption (Fig. 3f, g). Similarly as $Mir182^{\Delta M/\Delta M}$ mice, the CH-nanoparticles containing miR-182 inhibitor oligos did not prominently affect joint swelling (Supplementary Fig. 10), indicating that miR-182 inhibition does not significantly suppress immune response in this arthritis model. Thus, inhibition of miR-182 could serve as a promising complementary approach to control excessive osteoclastogenesis and bone loss in inflammatory settings while minimizing undesired side effects on immune suppression.

Taken together of both of the genetic and pharmacological evidence, miR-182 plays a prominent role in promoting osteoclastic bone resorption in pathological conditions. Inhibition of miR-182 reveals its therapeutic benefits against pathologic bone loss, such as that occurs in osteoporosis and inflammatory arthritis.

**miR-182 suppresses autocrine IFN-β pathway by targeting PKR.** We next sought to investigate the mechanisms by which miR-182 functions as an osteoclastogenic regulator. To address this question, we first performed gene expression profiling using high-throughput sequencing of RNAs (RNA-seq) with the WT control and $Mir182^{\Delta M/\Delta M}$ osteoclast precursors at baseline and after RANKL stimulation to identify genes regulated by miR-182 during osteoclastogenesis. In this study, two biological RNA-seq replicates were performed for each condition followed by Pearson correlation analysis to assess the reproducibility of RNA-seq data. As shown in Supplementary Fig. 11, the gene expression values of the two independent biological RNA-seq replicates are highly correlated for each condition with Pearson's $R \geq 0.986$, indicating a markedly high reproducibility between these replicates for each condition. We then analyzed the RNA-seq data using these replicates. RNA-seq-based expression heatmap of osteoclastic genes regulated by miR-182 showed that the lack of miR-182 significantly suppressed the expression of osteoclastogenic regulators, such as NFATc1 and Prdm1, and osteoclast marker genes, such as TRAP, cathepsin K, OCSTAMP, DCSTAMP, and Atp6v0d2 (Fig. 4a). These RNA-seq data are in alignment with our findings on the suppressed osteoclastogenesis phenotype resulted from miR-182 deficiency (Fig. 1). Gene Ontology (GO) analysis of the upregulated genes by miR-182 deficiency during RANKL-induced osteoclastogenesis revealed highly significant

activation of genes involved in defense response, in particular interferon-β response ($p < 10^{-25}$, Fig. 4b). Gene set enrichment analysis (GSEA) of the miR-182 regulated genes also revealed the top significant enrichment of type I interferons (IFN) response genes ($p < 0.0001$ and FDR $< 0.0001$) in $Mir182^{\Delta M/\Delta M}$ cells in response to RANKL (Fig. 4c). We further extracted the gene expression values of the most enriched type I IFN response gene set from our RNA-seq data and confirmed 52 RANKL-regulated IFN response genes that were highly enhanced in $Mir182^{\Delta M/\Delta M}$ cells, such as MX1, IFIT2, IRF7, and CXCL10 (Fig. 4d). Previous literature demonstrates that interferon pathway (IFN α, β, or γ) inhibits osteoclast differentiation[42,44,51,52]. The endogenous IFN-β induction by RANKL in osteoclasts is an important inhibitory feedback loop to maintain a balanced differentiation state[44,52]. Thus, the highly significant enrichment of type I IFN response gene sets in the $Mir182^{\Delta M/\Delta M}$ cells reflects a significantly augmented IFN pathway and could explain the phenotype of the suppressed osteoclastogenesis by miR-182 absence. It is of interest how a miRNA controls such a specifically enriched group of genes. It is unlikely that miR-182 directly targets all of these genes because most of them do not contain miR-182 seed region. We then hypothesized that miR-182 directly targets certain genes that in turn specifically regulate IFN responsive genes during osteoclastogenesis. To this end, we performed two parallel RNA-seq experiments, one with miR-182 loss of function approach using the WT and $Mir182^{\Delta M/\Delta M}$ BMMs and the other with miR-182 gain of function approach using the control and $Mir182^{mTg}$ BMMs, to first identify genes regulated by miR-182 in osteoclast precursors. We first performed differential expression analysis using edgeR and selected genes that were upregulated at least 1.2-fold by the miR-182 deficiency and suppressed at least 1.2-fold by the miR-182 overexpression with FDR $< 0.05$. With this complementary approach we obtained a list of 24 genes, which contained miR-182 potential targets. We then overlapped these genes with a published database that has experimentally verified miR-182 targets[27], and determined four miR-182 direct targets (Fig. 4e). In this group of genes, the gene $Eif2ak2$ encoding protein kinase double-stranded RNA-dependent (PKR) stood out with the highest expression level, whereas the other three genes were only marginally expressed (Fig. 4e). Importantly, among these four genes, only PKR is a well characterized factor that plays a critical role in type I IFN pathway[53–55], which indeed also fit our hypothesis that miR-182 targets PKR that further regulates downstream IFN pathway. We further confirmed the seed region of miR-182 in the 3′ UTR of $Eif2ak2$ (Fig. 4f), and validated the regulation of both mRNA and protein expression levels of PKR by miR-182 using both $Mir182^{\Delta M/\Delta M}$ BMMs and $Mir182^{mTg}$ BMMs (Fig. 4g, h). Furthermore, we performed a 3′ UTR

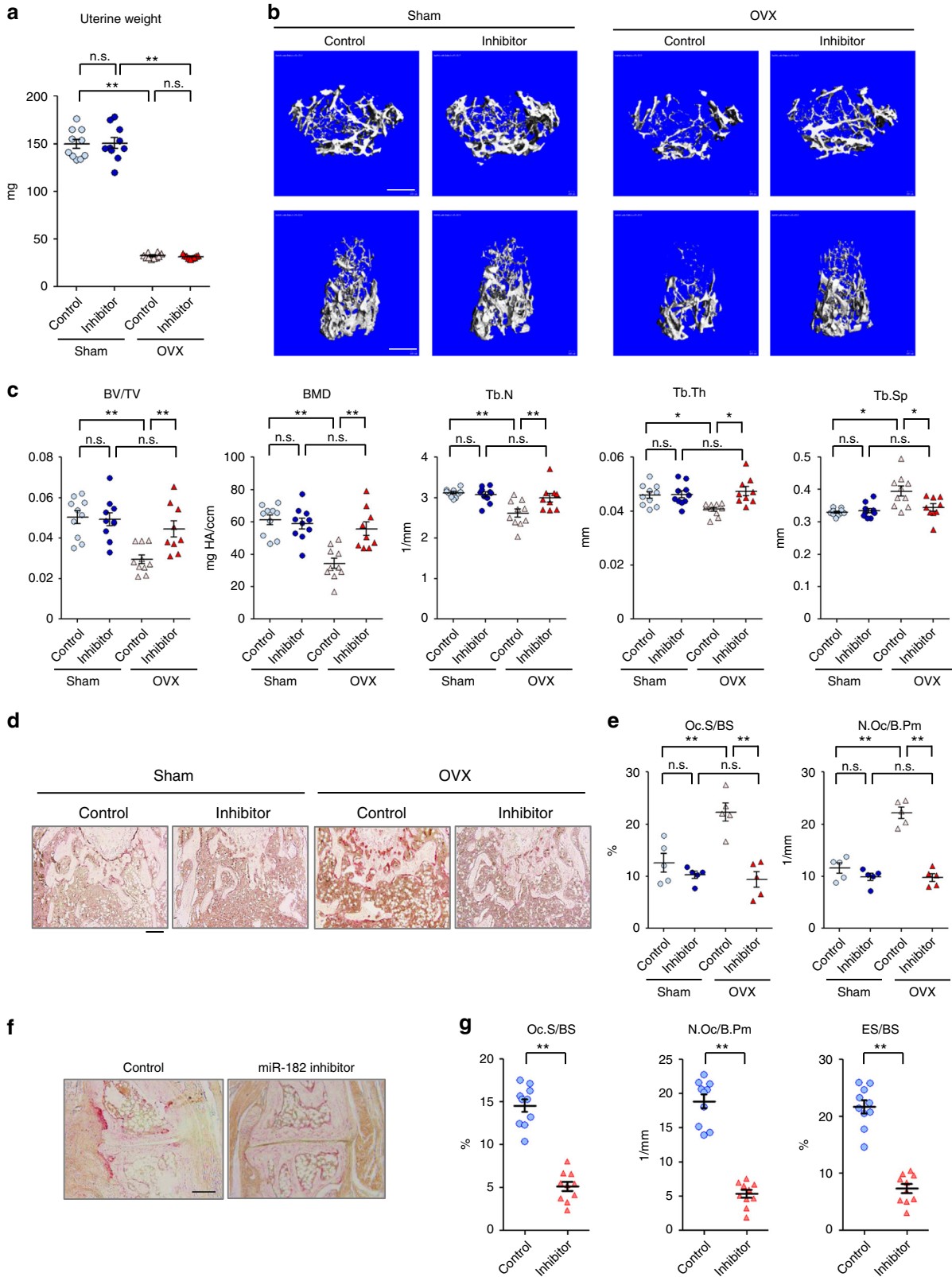

luciferase reporter assay, and found that miR-182 mimic down-regulated the luciferase activity of the 3′ UTR of *Eif2ak2* reporter (Fig. 4i). Mutation of the seed region of miR-182 in the 3′ UTR of *Eif2ak2* abolished the regulatory effect of miR-182 on the 3′ UTR of *Eif2ak2* (Fig. 4f, i). These results collectively validate that PKR is directly targeted by miR-182.

Following this line, we examined the role of PKR in osteoclast differentiation, which has not been studied. The expression of PKR was decreased along osteoclast differentiation (Fig. 5c, Supplementary Fig. 12), indicating that PKR might be a negative regulator in this process. To address this question, we utilized the BMMs isolated from the PKR knockout (*Pkr*−/−) and WT control

**Fig. 3** Pharmacological inhibition of miR-182 suppresses pathologic bone resorption in disease models. **a–e** OVX or sham operation was performed on 10-week-old female C57BL/6 mice. Three days after surgery, the OVX and sham mice were treated with Control or miR182 inhibitor-carrying CH-nanoparticles ($n = 10$ in the groups of Sham-control, Sham-miR-182 Inhibitor and OVX-control, and $n = 9$ in the group of OVX-miR-182 Inhibitor) at 5 μg per mouse twice a week for 5 weeks. **a** Uterine weight, **b** μCT images, and **c** bone morphometric analysis of trabecular bone of the distal femurs isolated from each group. **d** TRAP staining and **e** histomorphometric analysis of histological sections obtained from the metaphysis region of distal femurs isolated from the indicated mice. $n = 5$ per group. **f–g** Inflammatory arthritis was induced by intraperitoneal injection of 100 μl of K/BxN serum to the mice on days 0 and 2. Control or miR182 inhibitor-carrying CH-nanoparticles were delivered by intravenous injections at 5 μg on day 2, 4, 6, and 8. Mice were sacrificed on day 10. **f** TRAP staining and **g** histomorphometric analysis of histological sections of tarsal joints obtained from each group ($n = 10$ per group). BV/TV, bone volume per tissue volume; BMD, bone mineral density; Tb.N, trabecular number; Tb.Th, trabecular thickness; Tb.Sp, trabecular separation; Oc.S/BS, osteoclast surface per bone surface; N.Oc/B.Pm, number of osteoclasts per bone perimeter; ES/BS, erosion surface per bone surface. **a, c, e** $^*p < 0.05$; $^{**}p < 0.01$; n.s., not statistically significant by two-way ANOVA. **g** $^{**}p < 0.01$ by Student's $t$ test. Error bars: data are mean ± SEM. Scale bars: **b** 500 μm; **d, f** 200 μm

mice ($Pkr^{+/+}$) as osteoclast precursors in three osteoclastic culture systems activated by RANKL alone, RANKL or TNF priming (Fig. 5a). In all conditions, PKR deficiency dramatically increased osteoclast differentiation assessed by enhanced osteoclast generation and elevated gene expressions of osteoclastogenic transcription factors, such as NFATc1 and Blimp1 (Fig. 5b, c, Supplementary Fig. 12), as well as of osteoclast markers, such as Ctsk and Dcstamp (Fig. 5b). These results indicate that PKR is a novel inhibitor of osteoclastogenesis. Due to PKR's involvement in type I IFN pathway especially as an activator of IFN-β expression[53,54], we examined its expression and found that PKR deficiency almost completely blocked the induction of IFN-β by RANKL (Fig. 5d). As a consequence, the induction of IFN-β response genes, such as $Mx1$, $Ifit1$, and $Irf7$, was strongly suppressed in the PKR deficient cell cultures induced by RANKL relative to the control cell cultures (Fig. 5e). Next, we tested the functional importance of the PKR-regulated IFN-β changes in osteoclastogenesis. We blocked the inhibitory function of endogenous IFN-β on osteoclastogenesis using an IFN-β neutralizing antibody. Consistent with previous findings, blocking of endogenous IFN-β enhanced RANKL-induced osteoclastogenesis in WT cell cultures (Fig. 5f). The effect of PKR deficiency on the enhanced osteoclastogenesis, however, was abrogated by blocking IFN-β, as evidenced by the data that PKR deletion increased osteoclastogenesis in the presence of the control IgG, but not in the presence of the IFN-β neutralizing antibody (Fig. 5f). These results demonstrate that PKR is a new negative regulator of osteoclastogenesis, which exerts inhibitory function via induction of endogenous IFN-β expression. Finally, we knocked down PKR expression in the $Mir182^{\Delta M/\Delta M}$ BMMs (Fig. 5g), and found that the suppressed osteoclast differentiation in $Mir182^{\Delta M/\Delta M}$ BMMs was significantly reversed to a level similar to that in the WT cells (Fig. 5h). These results indicate that PKR is an essential downstream target responsible for the miR-182 function in osteoclastogenesis. To further test whether IFN-β is a critical downstream effector of miR-182, we examined IFN-β expression and found that miR-182 deficiency significantly enhanced the RANKL-induced IFN-β levels (Fig. 6a). Importantly, the protein levels of IFN-β in the serum were also markedly increased in the $Mir182^{\Delta M/\Delta M}$ mice (Fig. 6b). Furthermore, blocking of endogenous IFN-β using an IFN-β neutralizing antibody (Fig. 6c) or by knocking down IFN-β expression (Fig. 6d, e) abrogated the inhibitory effect of miR-182 deficiency on the RANKL-induced osteoclastogenesis. Neutralization of endogenous IFN-β did not have additional effect on the miR-182 overexpression-enhanced osteoclastogenesis in the $Mir182^{mTg}$ cell cultures (Fig. 6f). These findings indicate that IFN-β is a critical downstream component of miR-182 during osteoclastogenesis. Collectively, our data identify a previously unrecognized mechanism (Fig. 6g) by which miR-182 directly targets PKR,

downregulation of which in turn suppresses endogenous IFN-β expression induced by RANKL, thereby tuning down the feedback inhibitory loop and activating osteoclastogenesis. This newly identified miR-182-PKR-IFN-β pathway thus plays an essential role in osteoclast differentiation and osteoclastogenic gene expression program.

**miR-182-PKR-IFN-β axis is significantly correlated with RA.** Finally we set out to examine the expression levels of miR-182 in the human osteoclast precursor CD14 (+) PBMCs isolated from healthy donors and RA patients. As shown in Fig. 7a, human miR-182 level was drastically elevated in RA patients. Given the importance of TNF-α in the pathogenesis of RA and the resounding success of TNF blockade therapy (TNFi) in the treatment of RA, we next stimulated human CD14 (+) PBMCs with TNF-α and found that TNF-α induced miR-182 expression (Fig. 7b). TNF blockade therapy of RA patients with a humanized antibody (Enbrel) that specifically blocks TNF activity strikingly decreased miR-182 expression levels in CD14 (+) PBMCs after one or two month-treatment (Fig. 7c left panel). In parallel, TNFi significantly decreased the osteoclastogenesis of RA PBMCs (Fig. 7c right panel). Moreover, the therapeutic reduction of TNF activity suppressed miR-182 expression as well as osteoclastogenesis in each individual RA patient after TNFi therapy (Fig. 7d). Further statistical analysis revealed a strong correlation (Pearson's $r = 0.748$) between the levels of miR-182 expression and osteoclastogenesis in RA PBMCs (Fig. 7e). We next determined the expression levels of PKR and IFN-β, the important downstream factors of miR-182. In contrast to the higher level of miR-182 in RA (Fig. 7a), both PKR and IFN-β levels were markedly lower in RA PBMCs than in healthy donor cells (Fig. 7f). TNFi therapy elevated the expression levels of PKR and IFN-β in RA PBMCs (Fig. 7g). Significant elevation of these two genes was also determined in the monocytes isolated from each individual RA patient after TNFi treatment (Fig. 7h). The osteoclastic levels of PBMCs exhibited a strong negative correlation with the expression levels of PKR and IFN-β (Pearson's $r = -0.611$ and $-0.686$, respectively, in Fig. 7i). These results clearly show that the regulatory pattern of the miR-182-PKR-IFN-β axis is well conserved in RA PBMCs. Upregulation of miR-182 accompanying with the downregulation of PKR and IFN-β is strongly correlated with osteoclastogenic levels in RA. The striking correlation between the miR-182-PKR-IFN-β axis and osteoclastogenic potential of RA PBMCs provides a piece of evidence that supports the osteoclastogenic function of miR-182 in humans (Fig. 1e, f). Taken together with the proof-of-concept therapeutic benefits obtained from animal models (Figs. 2, 3), these human data highlight a promising therapeutic implication of targeting miR-182 in suppressing excessive osteoclastogenesis and bone erosion in human diseases.

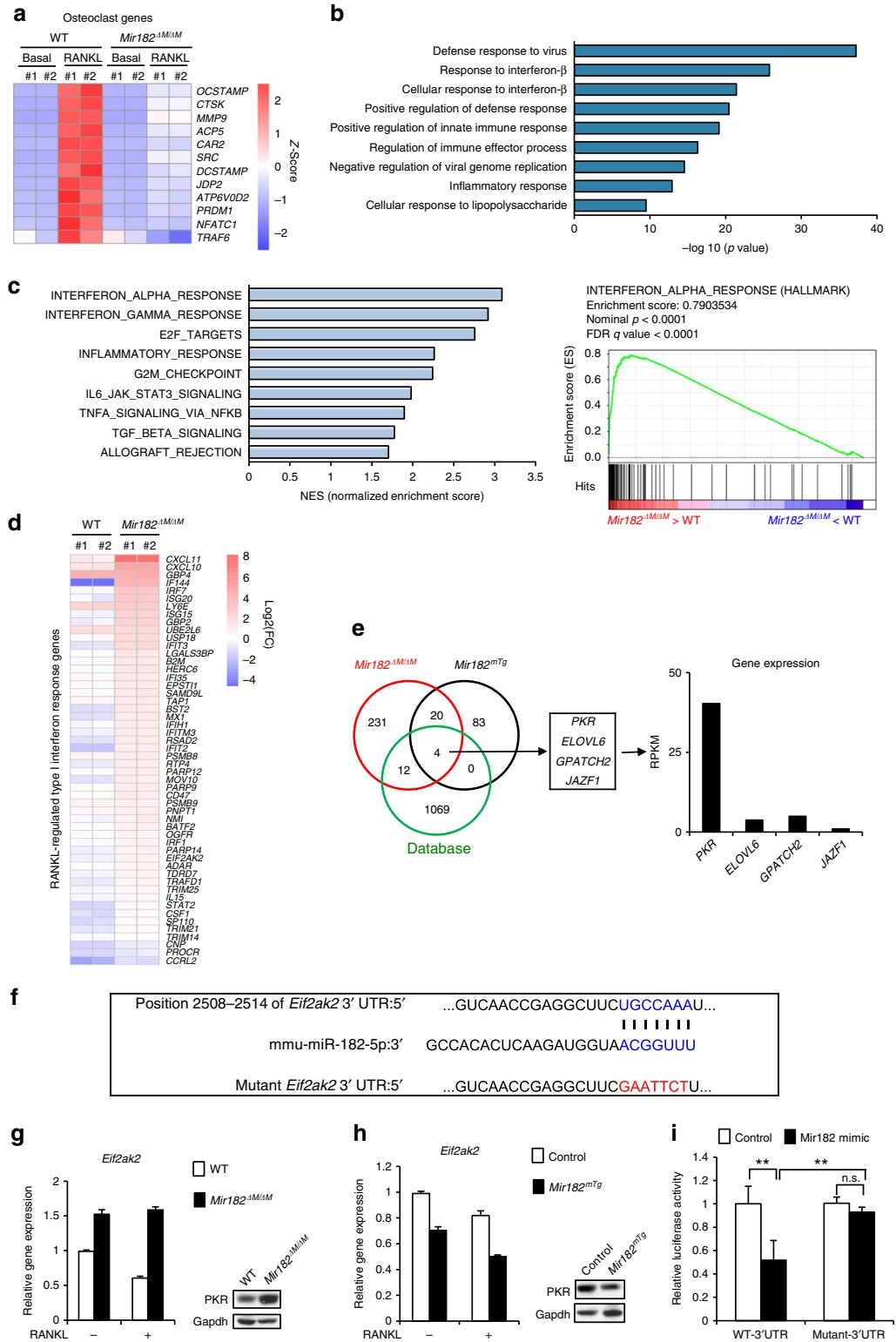

## Discussion

miRNAs exert functions through their specific targets and the downstream pathways mediated by the targets. Identification of miRNA targets can reveal molecular mechanisms and regulatory networks of each miRNA. By binding to complementary seed region in target mRNAs, different miRNAs target different genes. The target genes regulated by the same miRNA can also be variable depending on cell and tissue types as well as various stresses or disease settings[7,9–11,22,23,30], presumably due to diverse gene expression and regulation profiles in different conditions. In this study, we identified *eIF2ak2* (encoding PKR) as a new target for miR-182 in response to RANKL during osteoclasogenesis. Although other targets have been reported in different contexts[25–34], PKR is an essential target of miR-182 in RANKL-induced osteoclast differentiation evidenced by the data that genetic deletion of PKR abolishes the function of miR-182 in

**Fig. 4** miR-182 inhibits endogenous IFN-β signaling via targeting PKR. **a**–**d** RNA sequencing was performed using the mRNAs from the WT and $Mir182^{\Delta M/\Delta M}$ BMMs stimulated with or without RANKL for five days. $n = 2$ for each condition. #1, replicate 1. #2, replicate 2. **a** Heatmap of RANKL-induced osteoclast marker genes and transcription factors regulated by miR-182 deficiency. Row z-scores of CPMs of osteoclast genes were shown in the heatmap. **b** Gene ontology analysis of RANKL-inducible genes regulated by miR-182 deficiency. **c** Gene set enrichment analysis of RANKL-inducible genes regulated by miR-182 deficiency ranked by NES. The enrichment plot of the hallmark of type I IFN response was shown on the right panel. **d** Heatmap of the enriched RANKL-regulated type I interferon-response genes enhanced by miR-182 deficiency. #1, replicate 1. #2, replicate 2. **e** Left, Venn diagram showing the overlap of the down-regulated genes in the $Mir182^{mTg}$ BMMs, upregulated genes in the $Mir182^{\Delta M/\Delta M}$ BMMs and the miR-182 targets obtained from a database[27]. RNA sequencing data obtained from the mRNAs from the WT and $Mir182^{\Delta M/\Delta M}$ BMMs, or the Control and $Mir182^{mTg}$ BMMs were used. $n = 2$ for each condition. Middle, a gene list extracted from the overlap of the three gene sets. Right, RPKM based gene expression of the four indicated genes. **f** Seed region of miR-182 in the 3′ untranslated region of mouse $Eif2ak2$. **g** Quantitative real-time PCR (qPCR) analysis of mRNA expression of $Eif2ak2$ (encoding PKR) from WT and $Mir182^{\Delta M/\Delta M}$ BMMs treated with or without RANKL for five days. Right, immunoblot analysis of PKR expression in WT and $Mir182^{\Delta M/\Delta M}$ BMMs. **h** qPCR analysis of mRNA expression of $Eif2ak2$ from the Control and $Mir182^{mTg}$ BMMs treated with or without RANKL for two days. Right, immunoblot analysis of PKR expression in the control and $Mir182^{mTg}$ BMMs. **g**, **h** Gapdh was used as a loading control. **i** Luciferase activities measured from the HEK293 cells co-transfected with the WT or mutant 3′ UTR of $Eif2ak2$ luciferase reporter plasmids together with miR-182 mimic or the corresponding control ($n = 3$). Data are mean ± SD. **$p < 0.01$; n.s. not statistically significant by two-way ANOVA

osteoclastogenesis. PKR was initially discovered as an IFN-β inducible gene in the antiviral response[53–57]. Double-stranded RNA generated during viral replication activates PKR, which functions as a kinase that phosphorylates the α-subunit of eukaryotic translation initiation factor 2 (eIF2α) on Ser51. Phosphorylation of eIF2α blocks translational initiation and thereby promotes cellular apoptosis. PKR plays an important role in type I IFN production. Recently, PKR has also been implicated in regulating other cellular functions such as cell growth and differentiation, gene transcription and translation, and signal transduction. In addition to viruses, PKR can be activated by other stimulations, such as TLRs, TNF, Insulin, and ER stress[58–61]. The downstream signaling pathways activated by PKR in these conditions are diverse, such as IFN-β, NF-κB signaling, MAPK pathways and eIF2a-mediated translational inhibition, dependent on cell types and context[58–61]. At the early stage of osteoclast differentiation, PKR could presumably be regulated directly by RANKL or indirectly by the autocrine IFN-β induced by RANKL. The significant downstream factor mediated by PKR in osteoclastogenesis is IFN-β, which is supported by our data that PKR deletion almost abrogates RANKL-induced expression of IFN-β and the downstream type I IFN response genes. Along differentiation, RANKL induces the expression of miR-182, which in turn downregulates PKR expression, eventually shutting off IFN-β expression and allowing osteoclastogenesis to proceed efficiently. Few miRNAs were reported to act directly on PKR. miR-182 thus is an miRNA newly identified to directly target PKR, introducing miR-182 as an important regulator for IFN-β signaling pathway.

Physiological osteoclast differentiation is delicately controlled and maintained by complex mechanisms at various levels. The extent of osteoclastogenesis is determined by the balance between osteoclastogenic and anti-osteoclastogenic factors. Endogenous IFN-β (even at very low levels) produced by macrophages/osteoclast precursors in response to RANKL is a strong feedback mechanism that inhibits osteoclast differentiation[44,52]. RANK signaling needs to overcome the negative regulatory mechanisms, such as by downregulating expression of transcriptional repressors, to allow osteoclastogenesis to proceed[52]. Previously it was unclear how the IFN-β mediated inhibitory loop was tuned down during osteoclast differentiation. Our studies uncovered an important mechanism that the miR-182-PKR axis is responsible for suppressing autocrine IFN-β signaling. Kinetically, we observed that miR-182 appears to be induced at a later time than IFN-β during osteoclastogenesis. Thus, in order for osteoclastogenesis to proceed smoothly, RANKL induces a key positive regulator miR-182 to specifically tune down a negative pathway

mediated by IFN-β that is activated at the early stage of differentiation. These findings add a new regulatory circuit orchestrated by miR-182 to the positive and negative regulatory network of osteoclastogenesis (Fig. 6g). Overexpression of miR-182 enables RANKL to induce efficient osteoclast differentiation at lower concentrations with approximately 25–50% of those used in the control cell cultures (Supplementary Fig. 2d). On the other hand, much higher concentrations of RANKL are required to induce osteoclastogenesis in the miR-182 deficient cells (Supplementary Fig. 1). These results support the finding that miR-182 functions as a RANKL-inducible feed forward regulator to promote osteoclast differentiation. This regulatory mechanism is a part of downstream signaling cascades of RANKL. Although the strategies of direct blocking RANK receptor or neutralizing RANKL are very effective in inhibition of osteoclastogenesis, they could strongly or completely shut off osteoclastogenic program, leading to potential long-term side effects, such as inhibition of osteoclast-mediated bone remodeling. It is thus beneficial to develop complementary approaches to control abnormal osteoclastogenesis in disease conditions. Pathological scenarios, such as inflammatory conditions in RA, can activate the miR-182-PKR pathway in synergy with RANKL to promote osteoclast differentiation. Therefore, appropriate regulation of miR-182-PKR axis provides alternative possibilities to fine tune osteoclastogenesis instead of complete shutting off differentiation program in disease settings. Indeed, the data from our therapeutic models show that the pharmacological inhibition of miR-182 suppresses excessive osteoclastogenesis, but does not completely block the differentiation and bone remodeling. Our findings provide new insights into mechanisms that control the balance between positive and negative pathways that determine the extent of osteoclastogenesis and identify new therapeutic targets for inhibition of pathologic bone resorption.

Despite a significant progress in miRNA therapeutics, only a small number of miRNA mimics or inhibitors have entered clinical development[15,16]. One challenge is the design of miRNA delivery approaches that can ideally make the miRNA-based drugs stable and enable tissue-specific targeting, meanwhile minimizing potential toxicities and off-target effects. Naked small RNA molecules are easily degradable. Chemical modification of the nucleotide backbone of miRNA mimics or inhibitors, such as modification with locked nucleic acid (LNA), have improved their binding affinity and stability[15]. In our studies, the miR-182 inhibitor has LNA modification. The initial preliminary results, however, showed that a large amount of the miR-182 inhibitors (~1 mg daily) was required to suppress osteoclastogenesis in vivo, indicating a delivery vehicle is necessary to reduce amount.

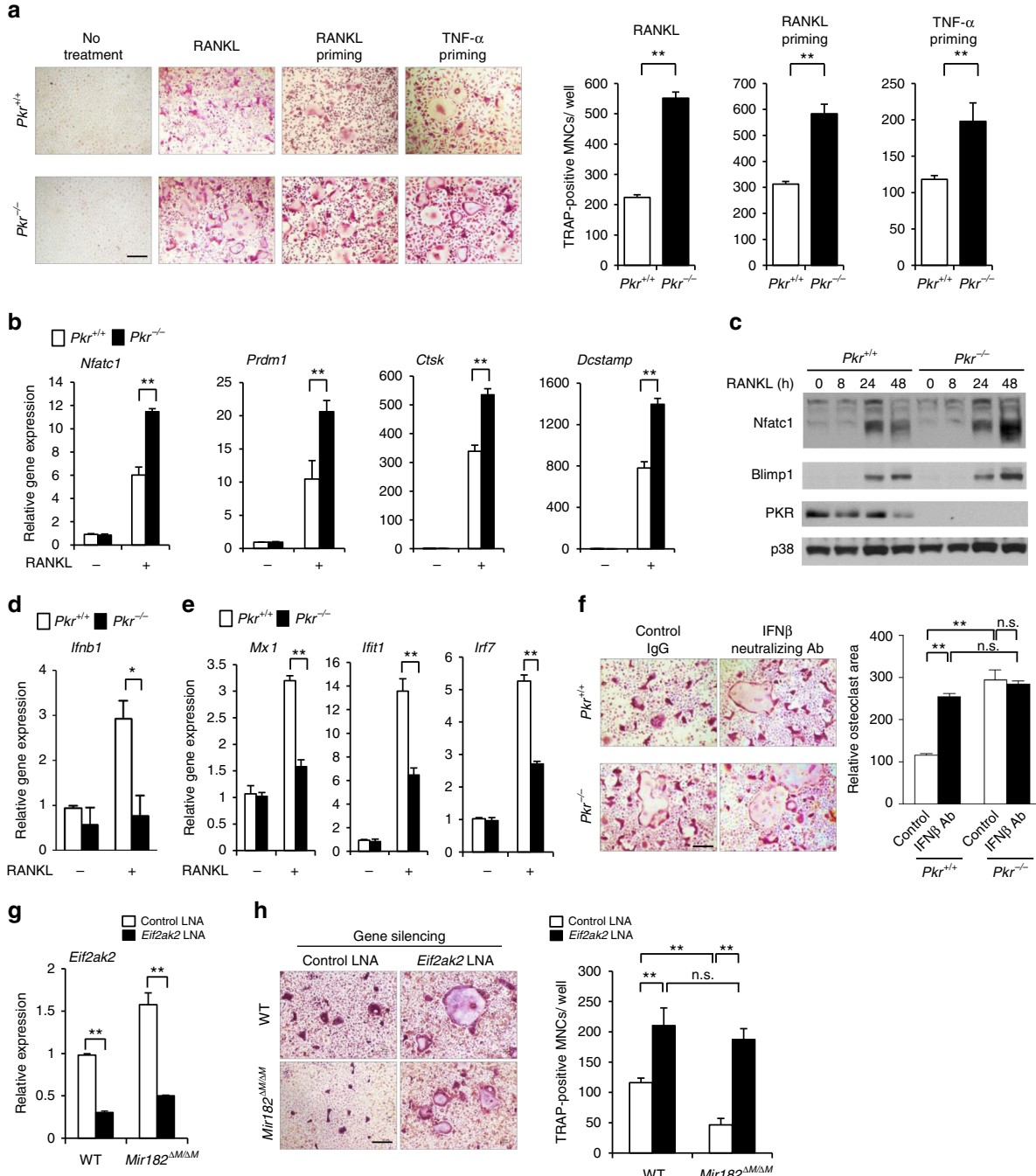

**Fig. 5** PKR as an essential target of miR-182 suppresses osteoclastogenesis by regulating IFN-β feedback. **a** Osteoclast differentiation determined by TRAP staining (left panel) and the relative area of TRAP-positive MNCs per well (right panel) in the cell cultures using the $Pkr^{+/+}$ and $Pkr^{-/-}$ BMMs that were stimulated with RANKL for 3 days, or primed with TNF-α or RANKL for one day followed by costimulation with TNF-α and RANKL for 2 days. TRAP-positive cells appear red in the photographs. **b** Quantitative real-time PCR analysis of mRNA expression of *Nfatc1*, *Prdm1*, *Ctsk*, and *Dcstamp* in BMMs from $Pkr^{+/+}$ and $Pkr^{-/-}$ mice treated with or without RANKL for two days. **c** Immunoblot analysis of the expression of NFATc1, Blimp1, and PKR induced by RANKL at the indicated times. p38 was used as a loading control. **d**, **e** Quantitative real-time PCR analysis of mRNA expression of *Ifnb1* (**d**), *Mx1*, *Ifit1*, and *Irf7* (**e**) in $Pkr^{+/+}$ and $Pkr^{-/-}$ BMMs treated with or without RANKL for two days. **f** BMMs from $Pkr^{+/+}$ and $Pkr^{-/-}$ mice were treated with control IgG (10 U/ml) or IFNβ-neutralizing Ab (10 U/ml) in the presence of RANKL for 3 days. Left panel: TRAP staining; Right panel: quantification of the relative area of TRAP + MNCs per well. **g** Quantitative real-time PCR analysis of mRNA expression of *Eif2ak2* showing the knockdown efficiency of PKR in the WT and $Mir182^{\Delta M/\Delta M}$ BMMs that were transfected with the control or *Eif2ak2* specific LNAs (40 nM). **h** Osteoclast differentiation determined by TRAP staining (left panel) and the relative area of TRAP-positive MNCs per well (right panel) in the WT and $Mir182^{\Delta M/\Delta M}$ BMM cell cultures transfected with the control or *Eif2ak2* specific LNAs and stimulated with RANKL for 3 days. **a**, **b**, **d**, **g** $^{**}p < 0.01$ by Student's *t* test. **f**, **h** $^{**}p < 0.01$; n.s. not statistically significant by two-way ANOVA. Error bars: data are mean ± SD. Scale bars: **a**, **f**, **h** 200 μm

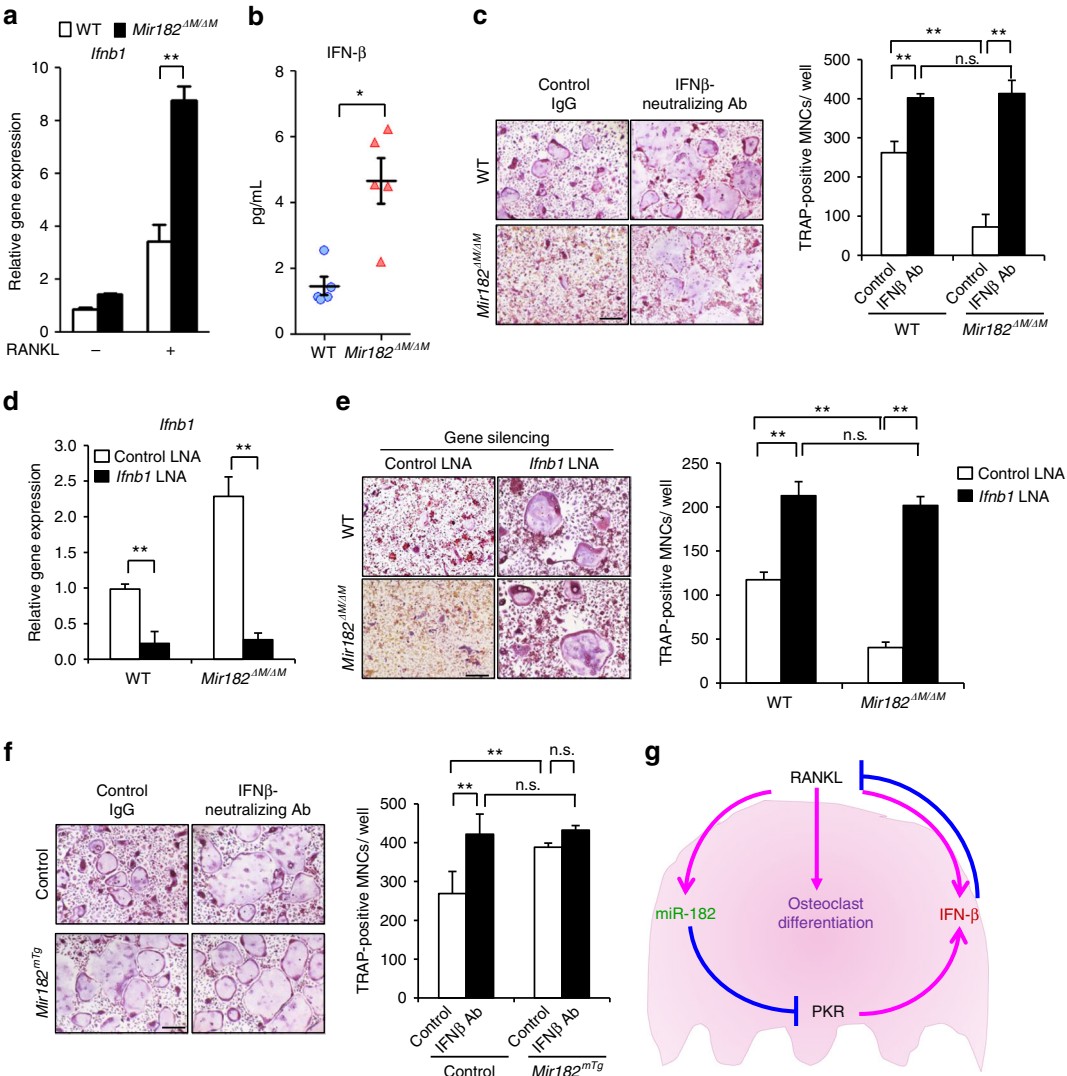

**Fig. 6** IFN-β is a critical downstream effector of miR-182. **a** Quantitative real-time PCR analysis of mRNA expression of *Ifnb1* in the WT and *Mir182^{ΔM/ΔM}* BMMs treated with or without RANKL for 2 days. **b** ELISA analysis of the serum IFN-β levels in the WT and *Mir182^{ΔM/ΔM}* mice. $n = 5$ in each group. **c** Osteoclast differentiation using BMMs from WT and *Mir182^{ΔM/ΔM}* mice treated with control IgG (10 U/ml) or IFN-β-neutralizing antibody (10 U/ml) in the presence of RANKL for 3 days. Left panel: TRAP staining; Right panel: quantification of the relative area of TRAP + MNCs per well. **d** Quantitative real-time PCR analysis of mRNA expression of *Ifnb1* showing the knockdown efficiency of *Ifnb1* in the WT and *Mir182^{ΔM/ΔM}* BMMs that were transfected with the control or *Ifnb1* specific LNAs (40 nM). **e** Osteoclast differentiation determined by TRAP staining (left panel) and the relative area of TRAP-positive MNCs per well (right panel) in the WT and *Mir182^{ΔM/ΔM}* BMM cell cultures transfected with the control or *Ifnb1* specific LNAs and stimulated with RANKL for three days. **f** Osteoclast differentiation using BMMs from the Control and *Mir182^{mTg}* mice treated with control IgG (10 U/ml) or IFN-β-neutralizing antibody (10 U/ml) in the presence of RANKL for three days. Left panel: TRAP staining; Right panel: quantification of the relative area of TRAP + MNCs per well. **g** A model showing the newly identified regulatory circuit orchestrated by the miR-182-PKR-IFN-β axis, which mediates osteoclastogenesis. **a**, **b**, **d** *$p < 0.05$; **$p < 0.01$ by Student's *t* test. **c**, **e**, **f** **$p < 0.01$; n.s. not statistically significant by two-way ANOVA. Error bars: **a**, **c**–**f** Data are mean ± SD; **b** data are mean ± SEM. Scale bars: **c**, **e**, **f** 200 μm

Indeed, recent in vivo delivery technologies, including nanoparticle systems, have enabled the first generation of miRNA-based agents to move into the preclinical development and clinic trials[15,16]. CH is a cationic polymer derived from chitin and has been extensively used for small RNA delivery in preclinical studies due to its biodegradability and low cellular toxicity. We applied CH-nanoparticles to incorporate miR-182 inhibitors and reduced approximately 600 times of the amount of miR-182 inhibitors to 5 μg every 3 days to suppress osteoclastogenesis, and this small amount of miR-182 inhibitors showed efficient suppression of miR-182 expression in bone marrow (Supplementary Fig. 8e). The low amount of miR-182 inhibitors using CH delivery

system not only functions efficiently but also could lower off-targeting effect and cellular toxicity. CH is FDA-approved safe for wound dressing and dietary use, and there are several animal toxicity studies, including our results, reporting good safety in vivo[62]. Nonetheless, future studies in preclinical and clinical development need further optimization of the delivery system to eventually achieve a successful clinical application. The nanoparticle formula decides the particle size and weight that usually delicately determine the specificity of targeting certain cells[49,50]. For example, the CH formula used in this study facilitates targeting monocytes/macrophages and bone marrow. While our data from in vitro and in vivo experiments illustrate the

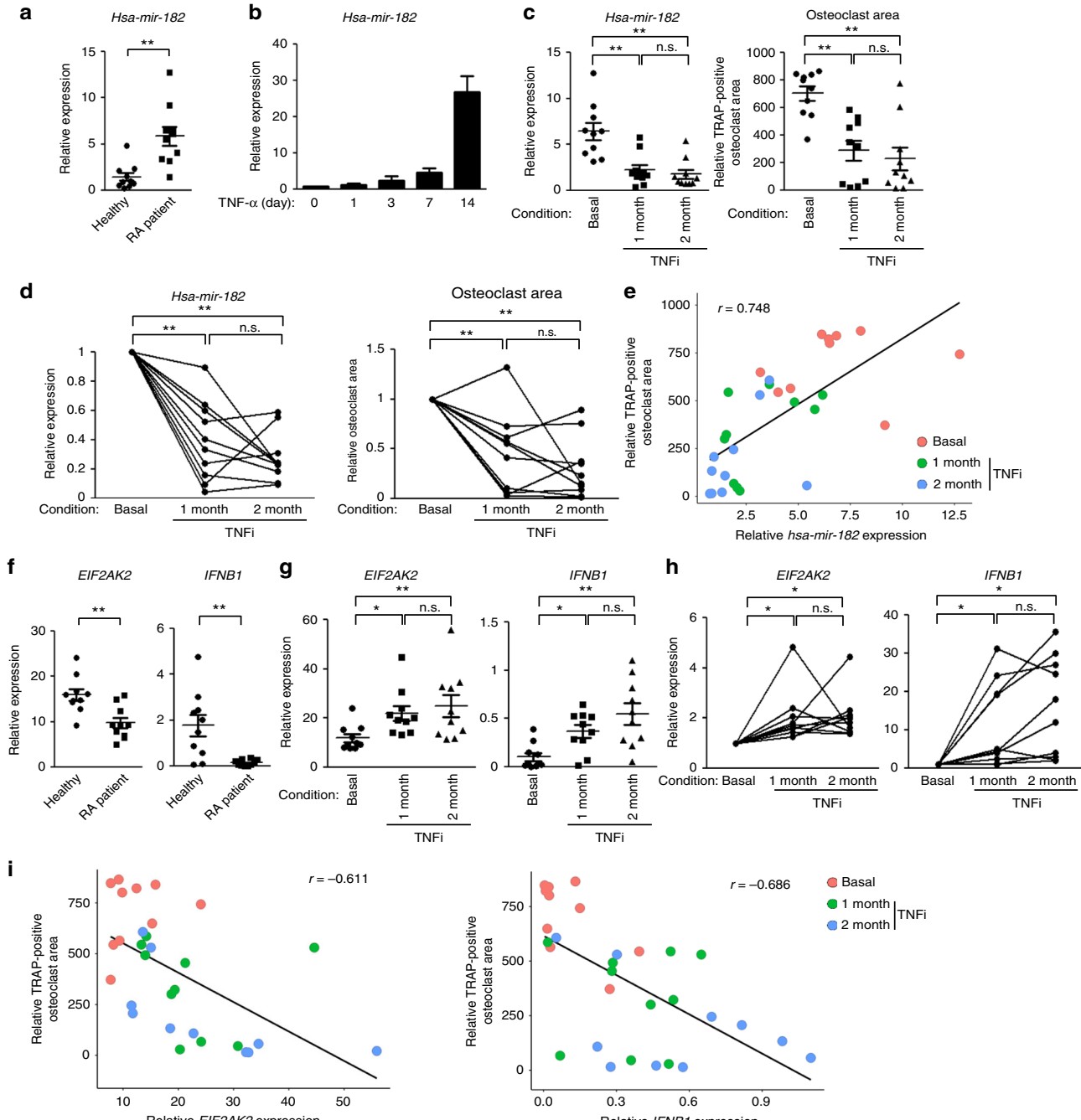

**Fig. 7** The expression of miR-182, PKR, and IFN-β is strongly correlated with RA osteoclastogenic capacity. **a** Quantitative real-time PCR (qPCR) analysis of *hsa-mir-182* expression in CD14(+) PBMCs isolated from healthy donors and RA patients. $n = 10$/group. **b** qPCR analysis of *hsa-mir-182* expression in human CD14(+) PBMC-derived macrophages treated with TNF-α (40 ng/ml). **c** Left, qPCR analysis of *hsa-mir-182* expression in CD14(+) PBMCs from RA patients before (basal) and after TNFi (Enbrel) for 1 and 2 months. Right, osteoclast differentiation indicated by the relative area of TRAP-positive MNCs in the cell cultures of human CD14(+) PBMC-derived macrophages treated with RANKL (100 ng/ml) for 5 days. Relative area of TRAP-positive MNCs was determined per well. $n = 10$/group. **d** Left, qPCR analysis of the relative expression of *hsa-mir-182* in CD14(+) PBMCs from the same RA patient before and after TNFi (Enbrel) for 1 and 2 months. Right, relative area of TRAP-positive MNCs. Results in each panel were normalized to the basal condition for each patient. $n = 10$/group. **e** Scatter plot showing significant positive correlation between the relative TRAP-positive osteoclast area and *hsa-mir-182* expression. Each dot represents an RA patient in the indicated conditions. Pearson's $r = 0.748$. **f** qPCR analysis of mRNA expression of *EIF2AK2* (encoding PKR) or *IFNB1* (encoding IFN-β) in CD14(+) PBMCs from healthy donors and RA patients. $n = 10$/group. **g** qPCR analysis of mRNA expression of *EIF2AK2* or *IFNB1* in CD14(+) PBMCs from RA patients before and after TNFi (Enbrel) for 1 and 2 months. $n = 10$/group. **h** Relative expression of *EIF2AK2* or *IFNB1* in CD14(+) PBMCs from the same RA patient before and after TNFi. Results were normalized to basal condition for each patient. $n = 10$/group. **i** Scatter plot showing significant negative correlation between the relative TRAP-positive osteoclast area and *EIF2AK2* (Left) or *IFNB1* (Right) expression. Each dot represents an RA patient. Pearson's $r = -0.611$ (left) or $-0.686$ (right). **a**, **f** **p < 0.01 by Student's *t* test; **c**, **d**, **g**, **h** **p < 0.01; n.s. not statistically significant by repeated measures ANOVA; data are mean ± SEM

inhibition of osteoclastogenesis and bone resorption by the miR-182 inhibitor, we are aware of the possibility that the miRNA inhibitor-containing nanoparticles might also target additional cells in vivo. Therefore, further analyses of other potential cellular targets of this miRNA inhibitor during long-term treatment in vivo as well as development of more osteoclast-specific targeting delivery approaches should be conducted when developing clinical therapeutic applications.

Our results show that miR-182 functions as a key osteoclastogenic factor in human osteoclastogenesis as well. The regulatory pattern of the expression changes of the miR-182-PKR-IFN-β axis along with osteoclastogenesis in mice is similarly determined in human cells, indicating that the miR-182-mediated mechanisms are likely to translate to human physiopathology. The positive correlation between miR-182 expression levels and the osteoclastogenic extent in humans also suggests a possibility of miR-182 as a clinical biomarker of osteoclastogenesis, which will require validation by large-scale cohort studies. Taken together of the genetic evidence from both in vitro and in vivo systems, strong correlation between miR-182 expression levels and osteoclastogenic potential in RA, and the in vivo pharmacological results obtained from animal disease models, our study provides a proof of concept for the efficacy of therapeutic targeting of miR-182 to prevent bone loss and highlights the translational implications of targeting miR-182 and its downstream targets in treating osteolytic diseases.

## Methods

**Animal study and analysis of bone phenotype**. We generated mice with myeloid-specific deletion of miR-182 by crossing *Mir182*[flox/flox] mice[30] with mice with a lysozyme M promoter-driven Cre transgene on the C57BL/6 background (known as *LysMcre*; The Jackson Laboratory). Gender- and age-matched *Mir182*[flox/flox]*LysMcre*(+) mice (referred to as *Mir182*[ΔM/ΔM]) and their littermates with *Mir182*[+/+]*LysMcre*(+) genotype as WT controls were used for experiments. Myeloid-specific miR-182 overexpression mice (referred to as *Mir182*[mTg]) were generated by crossing *LSL*(LoxP-Stop-LoxP)–*Mir182* mice[30], in which the mouse *Mir182* cDNA was knocked into the ubiquitously expressed *Rosa26* locus preceded by a *STOP* fragment flanked by *loxP* sites, and Cre activates miR-182 overexpression, with *LysMcre* on the C57BL/6 background (The Jackson Laboratory). Gender- and age-matched *Mir182*[mTg] mice and their littermates *LysMcre* mice as controls (referred to as Control) were used. *Eif2ak2* (encoding Pkr) knockout mice (referred to as *Pkr*[−/−]) were described previously[63]. The bone marrow isolated from gender- and age-matched *Pkr*[−/−] and WT *Pkr*[+/+] were used in the experiments. All animal procedures were approved by the Hospital for Special Surgery Institutional Animal Care and Use Committee (IACUC), and Weill Cornell Medical College IACUC.

Bilateral OVX or sham operation (Sham) was performed on 10-week-old female mice. The mice were sacrificed 5 weeks after surgery, uterine atrophy was first confirmed and then bones were collected for μCT and histological analysis.

For inflammatory osteolysis experiments, we used the established LPS-induced or TNF-induced supracalvarial osteolysis mouse model[42–46] with minor modifications. LPS was administrated at the dose of 12.5 mg/kg weight to the calvarial periosteum of age and gender-matched mice. Five days after LPS injection, the mice were sacrificed and the calvarial bones and serum were collected. For TNF-induced supracalvarial osteolysis model, TNF-α was administrated daily at the dose of 75 μg/kg to the calvarial periosteum of age and gender-matched mice for five consecutive days before the mice were sacrificed. The calvarial bones were subjected to μCT analysis, sectioning, TRAP staining, and histological analysis.

For Inflammatory arthritis experiments, we used K/BxN Serum Transfer-Induced Arthritis model[47]. K/BxN serum pools were prepared, and arthritis was induced by intraperitoneal injection of 100 μl of K/BxN serum to the female mice on days 0 and 2. The development of arthritis was monitored by measuring the thickness of wrist and ankle joints with digital slide caliper (Bel-Art Products). For each animal, joint thickness was calculated as the sum of the measurements of both wrists and both ankles. Joint thickness was represented as the average for each group. Mice were sacrificed on day 10 and serum and paws were collected. Hind paws were subjected to sectioning, TRAP staining, and histological analysis.

For in vivo miR-182 inhibitor delivery, we used CH-nanoparticles as a delivery vehicle[20]. The control or miR-182 inhibitor-carrying CH-nanoparticles were delivered by intravenous injections at 5 μg (in 100 μl PBS) per mouse, with twice a week for 5 weeks starting from 3 days after surgery for OVX models or on day 2, 4, 6, 8 for arthritis models. μCT analysis was conducted to evaluate bone volume and 3D bone architecture using a Scanco μCT-35 scanner (SCANCOMedical). Mice femora were fixed in 10% buffered formalin and scanned at 6 μm resolution.

Proximal femoral trabecular bone parameters or the cortical bone parameters obtained from the midshaft of femurs were analyzed using Scanco software according to the manufacturer's instructions and the American Society of Bone and Mineral Research (ASBMR) guidelines[43,64]. Femur and calvarial bones were subjected to sectioning, TRAP staining and histological analysis. For dynamic histomorphometric measures of bone formation, calcein (25 mg/kg, Sigma) was injected into mice intraperitoneally at 4 and 1 days before sacrifice to obtain double labeling of newly formed bones. The non-decalcified tibiae bones were embedded in methyl methacrylate. Sections measuring 5-μm thick were cut on a microtome (Leica RM2255, Leica Microsystems, Germany). For static histomorphometric measures of osteoblast parameters, undecalcified sections of the tibiae were stained using Masson-Goldner staining kit (MilliporeSigma). The osteomeasure software was used for bone histomorphometry using standard procedures according to the program's instruction.

**General experimental design**. Generally, sample sizes were calculated on the assumption that a 20–30% difference in the parameters measured would be considered biologically significant with an estimate of sigma of 10–20% of the expected mean (α set at 0.05). For each in vivo experiment, at least five mice per genotype or per treatment condition were used. No animals were excluded from the analysis. Experimental animals were grouped according to their genotypes. Within the same genotype group in each experiment, the mice were randomized to different procedures, such as Sham vs. OVX, or to different treatments, such as PBS vs. cytokine treatment, control nanoparticles vs. miR-182 inhibitor nanoparticles. Investigators performing μCT or histomorphometry analysis were blinded to the genotype and treatment group of each sample.

**Reagents**. Murine or human M-CSF, murine or human TNF-α, and soluble human RANKL were purchased from PeproTech. In vivo mouse miR-182 inhibitor and its in vivo corresponding control, *Eif2ak2* LNA, *Ifnb1* LNA, and negative-control LNA were purchased from Exiqon. Human miR-182 inhibitor and its control were purchased from Ambion. mirVana™ miRNA miR-182 mimic and its corresponding negative control (Cat # 4464058) were purchased from Life Technologies. Mouse rIFN-β was obtained from PBL Technology. IFN-β-neutralizing antibody (rabbit polyclonal antibody against mouse IFN-β) was from PBL Technology and the control IgG (rabbit) was obtained from Santa Cruz Biotechnology. The miR-182 inhibitor (5′-TTCTACCATTGCCAA-3′) or the corresponding control (5′-ACGTCTATACGCCCA-3′) (HPLC purified, Exiqon) was packaged into CH-nanoparticles[20]. The miR-182 inhibitor and its control obtained from Exiqon have fully phosphorothioate (PS)-modified backbones, which enhances stability, pharmacokinetic and pharmacodynamic properties in vivo (http://ipaper.ipapercms.dk/EXIQON/Marketing/guidelines/analyzing-rna-function-in-animal-models/).

**Cell culture**. To obtain BMMs, mouse bone marrow cells were harvested from tibiae and femora of age and gender-matched mutant and control mice and cultured for 3 days in α-MEM medium (Thermo Fisher Scientific) with 10% fetal bovine serum (FBS) (Atlanta Biologicals), glutamine (2.4 mM, Thermo Fisher Scientific), Penicillin–Streptomycin (Thermo Fisher Scientific) and CMG14–12 supernatant (condition medium, CM), which contained the equivalent of 20 ng/ml of rM-CSF and was used as a source of M-CSF[43]. The attached BMMs were scraped, seeded at a density of $4.5 \times 10^4/cm^2$, and cultured in α-MEM medium with 10% FBS, 1% glutamine and CM for overnight. Except where stated, the cells were then treated without or with optimized concentrations of RANKL (40 ng/ml) or TNF-α (40 ng/ml) in the presence of CM for times indicated in the figure legends. Culture media were exchanged every three days. For human osteoclast cultures, PBMCs from whole blood of healthy volunteers or RA patients were isolated by density gradient centrifugation using Ficoll (Invitrogen Life Technologies, Carlsbad, CA). CD14(+) cells were purified from fresh PBMCs using anti-CD14 magnetic beads (Miltenyi Biotec, Auburn, CA) as recommended by the manufacturer. Human CD14(+) monocytes were cultured in α-MEM medium with 10% FBS in the presence of M-CSF (20 ng/ml; PeproTech, Rocky Hill, NJ) for 2 days to obtain monocyte-derived macrophages, which were further cultured with RANKL for osteoclast differentiation. The RA CD14(+) PBMCs were from RA patients (age ≥18 and <70 years) who fulfilled American College of Rheumatology (ACR) 2010 RA classification criteria with disease duration <5 years and were under TNFi therapy for the first time (Enbrel, 25 mg weekly). Experiments with human cells were approved by Nanfang Hospital (China) and the Hospital for Special Surgery (USA) Institutional Review Board. Informed consent (PBMC collection) was obtained from all RA patients. TRAP staining was performed with an acid phosphatase leukocyte diagnostic kit (Sigma-Aldrich) in accordance with the manufacturer's instructions. Human embryonic kidney 293 cells were purchased from the American Type Culture Collection. HEK293 cells were grown in Dulbecco's modified Eagle's media with 10% FBS. The cell line was routinely tested for mycoplasma contamination.

**In vitro inhibition of miRNAs and gene silencing**. miRNA inhibitors specifically targeting *hsa-mir-182* or non-targeting control oligos (Ambion) were transfected into primary human CD14(+) monocytes at concentrations of 40 nM with the

Amaxa Nucleofector device set to program Y-001 using the Human Monocyte Nucleofector kit (Lonza). Antisense inhibition using locked nucleic acid (LNA) technology from Exiqon was applied to silence gene expression in vitro. LNA oligonucleotides specifically targeting *Eif2ak2* and non-targeting control LNAs were from Exiqon and were transfected into murine BMMs at concentrations of 40 nM using TransIT-TKO transfection reagent (Mirus) in accordance with the manufacturer's instructions.

**Reverse transcription and real-time PCR**. DNA-free RNA was obtained with the RNeasy MiniKit (Qiagen, Valencia, CA) with DNase treatment, and 1 μg of total RNA was reverse-transcribed with random hexamers and MMLV-Reverse Transcriptase (Thermo Fisher Scientific) according to the manufacturer's instructions. Real-time PCR was done in triplicate with the QuantStudio 5 Real-time PCR system and Fast SYBR® Green Master Mix (Thermo Fisher Scientific) with 500 nM primers. mRNA amounts were normalized relative to glyceraldehyde-3-phosphate dehydrogenase (GAPDH) mRNA. When RT was omitted, threshold cycle number increased by at least ten, signifying lack of genomic DNA contamination or nonspecific amplification; the generation of only the correct size amplification products was confirmed with agarose gel electrophoresis. The primers for real-time PCR were as follows: *Nfatc1*: 5′-CCCGTCACATTCTGGTCCAT-3′ and 5′-CAAG TAACCGTGTAGCTCCACAA-3′; *Prdm1*: 5′-TTCTTGTGTGGTATTGTCGGG ACTT-3′ and 5′-TTGGGGACACTCTTTGGGTAGAGTT-3′; Acp5: 5′-ACGGC TACTTGCGGTTTC-3′ and 5′-TCCTTGGGAGGCTGGTC-3′; *Ctsk*: 5′-AAGAT ATTGGTGGCTTTGG-3′ and 5′-ATCGCTGCGTCCCTCT-3′; *Itgb3*: 5′-CCGG GGGACTTAATGAGACCACTT-3′ and 5′-ACGCCCCAAATCCCACCCATA CA-3′; *Dc-stamp*: 5′-TTTGCCGCTGTGGACTATCTGC-3′ and 5′-AGACGTGG TTTAGGAATGCAGCTC-3′; *Pkr*: 5′-AACCCGGTGCCTCTTTATTC-3′ and 5′-ACTCCGGTCACGATTTGTTC-3′; *Ifnb1*: 5′-TTACACTGCCTTTGCCATCC-3′ and 5′-AGAAACACTGTCTGCTGGTG-3′; *Mx1*: 5′-GGCAGACACCACATAC AACC-3′ and 5′- CCTCAGGCTAGATGGCAAG-3′; *Ifit1*: 5′-CTCCACTTTCAG AGCCTTCG-3′ and 5′-TGCTGAGATGGACTGTGAGG-3′; *Irf7*: 5′-GTCTCGG CTTGTGCTTGTCT-3′ and 5′-CCAGGTCCATGAGGAAGTGT-3′; and *Gapdh*: 5′-ATCAAGAAGGTGGTGAAGCA-3′and 5′-AGACAACCTGGTCCTCAGTGT-3′; *NFATC1*: 5′-AAAGACGCAGAAACGACG-3′ and 5′-TCTCACTAACGGGA CATCAC-3′; *PRDM1*: 5′- TCTTGTGTGGTATTGTCGGGA-3′ and 5′-TGCTCG GTTGCTTTAGACTGC-3′; *CALCR*: 5′- CTGAAGCTTGAGCGCCTGAGTC-3′ and 5′-TGGGGTTGGGTGATTTAGAAGAAG-3′; *ITGB3*: 5′- GGAAGAACGCG CCAGAGCAAAATG-3′ and 5′-CCCCAAATCCCTCCCCACAAATAC-3′; *EIF2AK2*: 5′-AATGCCGCAGCCAAATTAGC-3′ and 5′-AGGCCTATGTAATT CCCCATGG-3′; *IFNB1*: 5′- AGAAGCTCCTGTGGCAATTG-3′ and 5′- ACTGCT GCAGCTGCTTAATC-3′; *GAPDH*: 5′-ATCAAGAAGGTGGTGAAGCA-3′ and 5′- GTCGCTGTTGAAGTCAGAGGA-3′.

For quantification of miRNA, total RNA was isolated, and the small RNA fraction was enriched with the mirVana miRNA Isolation Kit (Thermo Fisher Scientific) according to the manufacturer's instructions. For quantitative RT-PCR analysis of miRNA, cDNA was prepared from small RNAs with the TaqMan miRNA Reverse Transcription Kit (Thermo Fisher Scientific). TaqMan miRNA assays were used according to the manufacturer's recommendations (Thermo Fisher Scientific) for real-time PCR. The TaqMan U6 snRNA assay (Thermo Fisher Scientific) was used for normalization of miRNA expression values.

**Immunoblot analysis**. Total cell extracts were obtained using lysis buffer containing 150 mM Tris-HCl (pH 6.8), 6% SDS, 30% glycerol, and 0.03% Bromophenol Blue; 10% 2-ME was added immediately before harvesting cells. Cell lysates were fractionated on 7.5% SDS-PAGE, transferred to Immobilon-P membranes (Millipore), and incubated with specific antibodies. Western Lightning plus-ECL (PerkinElmer) was used for detection. NFATc1 antibody (556602, 1:1000) was from BD Biosciences; Blimp1 (sc-47732, 1:1000), PKR (sc-6282, 1:1000), GAPDH (sc-25778, 1:3000), and p38α (sc-535, 1:3000) antibodies were from Santa Cruz Biotechnology. Densitometry was performed using ImageJ software (http://imagej.nih.gov/ij/). The uncropped scans of the western blots are provided in the Supplementary information.

**RNA-seq and bioinformatics analysis**. Total RNA was extracted using RNeasy Mini Kit (QIAGEN) following the manufacturer's instructions. True-seq RNA Library preparation kits (Illumina) were used to purify poly-A + transcripts and generate libraries with multiplexed barcode adapters following the manufacturer's instructions. All samples passed quality control analysis using a Bioanalyzer 2100 (Agilent). RNA-seq libraries were constructed per the Illumina TrueSeq RNA sample preparation kit. High-throughput sequencing was performed using the Illumina HiSeq 4000 in the Weill Cornell Medical College Genomics Resources Core Facility. RNA-seq reads were aligned to the mouse genome (mm10) using TopHat[65] or HISAT2[66]. Cufflinks[67] was subsequently used to assemble the aligned reads into transcripts and then estimate the transcript abundances as RPKM (reads per kilo base per million) values. HT-seq[68] was used to calculate raw reads counts and edgeR[69] was used to calculate normalized counts as CPM (counts per million). Mapped reads were at an average of 39 million per library. Heatmaps were generated by pheatmap package in R. Genes with false discovery rate (FDR) < 0.05

were identified as significantly differentially expressed genes (DEG) between conditions using the edgeR analysis of two RNA-seq biological replicates. To identify miR-182 regulated pathways, we performed Gene Ontology (GO) analysis with Panther Classification System[70] input with the RANKL-regulated DEG genes that were more expressed in *Mir182*$^{\Delta M/\Delta M}$ cells than WT cells (≥1.5 fold). miR-182 regulated pathways by the GO analysis were ranked based on the *p* values. Gene Set Enrichment Analysis (GSEA program, Broad Institute) input with the RANKL-regulated DEG genes was performed according to the program's instructions. Hallmark pathways by GSEA were ranked based on the normalized enrichment score (NES). *p* and FDR values were calculated following the program's instructions.

**ELISA**. Mouse serum IFN-β was measured by using VeriKine-HS Mouse Interferon Beta Serum ELISA Kit (PBL Assay Science) according to the manufacturer's instruction.

**Luciferase reporter assay**. A 479 bp DNA fragment of the 3′ UTR of *Eif2ak2* containing the miR-182 seed sequence was cloned into the pMIR-REPORT miRNA Expression Reporter Vector (AM5795, ThermoFisher). The miR-182 seed sequence was mutated using PrimeSTAR MAX DNA polymerase (Takara BIO) to generate a mutant 3′UTR reporter plasmid with miss-matched miR-182 binding site. The 3′ UTR reporters were co-transfected to human embryonic kidney 293 cells with CMV-renilla luciferase reporter (as an internal transfection control), together with miR-182 mimic or a corresponding control using TransIT-TKO transfection reagent (Mirus) in accordance with the manufacturer's instructions. Forty-eight hours after transfection, the cells were lysed with passive lysis buffer (Promega), and firefly and renilla luciferase activities were measured using the dual-luciferase reporter assay system (Promega).

**Statistical analysis**. Statistical analysis was performed using Graphpad Prism® software. Two-tailed Student's *t* test was applied when there were only two groups of samples. In the case of more than two groups of samples, one-way ANOVA was used with one condition, and two-way ANOVA was used with more than two conditions. ANOVA analysis was followed by post hoc Bonferroni's correction for multiple comparisons. *p* < 0.05 was taken as statistically significant; *$^{*}$p* value < 0.05 and *$^{**}$p* value < 0.01. The data displayed normal distribution. The estimated variance was similar between experimental groups. Data are presented as the mean ± SD or ± SEM as indicated in the figure legends.

## Data availability

Data supporting the findings of this manuscript are available from the corresponding author upon reasonable request. RNA-seq data (accession #GSE108721) have been deposited in NCBI's Gene Expression Omnibus (http://www.ncbi.nlm.nih.gov/geo/query/acc.cgi?acc=GSE108721).

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

## Acknowledgments

We thank Drs. Gökhan Hotamisligil, Carl Nathan and Xiuju Jiang for providing $Pkr^{-/-}$ mice and their control mice, Dr. Giancarlo Chesi and David Kuo for technical assistance, and Dr. Steven R. Goldring for discussions. We are grateful to Christine Miller,

Mahmoud Elguindy, Gregory Vitone, Shin-ichi Nakano, and Cheng Xu from Dr. Bao-hong Zhao's laboratory for their helpful discussions and assistance. We thank Weill Cornell Genomics Resources Core Facility for their efficient and high quality sequencing service and related analysis. A.K.S. is supported by American Cancer Society Research Professor Award and NIH R35 CA209904, and L.Z. is supported by National Natural Science Foundation of China (31771051). This work was supported by grants from the National Institutes of Health (R00 AR062047, R01 AR068970, and R01 AR071463 to B. Z.). The content of this manuscript is solely the responsibilities of the authors and does not necessarily represent the official views of the NIH.

## Author contributions

K.I. designed and performed most of the experiments, analyzed data and contributed to manuscript preparation. Y.C. and L.Z. collected RA blood samples, isolated RA PBMCs and cultured human osteoclasts. Z.D., R.X., S.G., and M.B.G. assisted with experiments. E.G. contributed to RNA-seq analysis. D.G.K. provided *Mir182*$^{flox/flox}$ mice and *LSL–Mir182* mice. L.S.M., G.L.-B. and A.K.S. packaged and provided chitosan nano-particles containing miR-182 inhibitors or control oligos with formula optimization, bio-distribution, and quality control test. B.Z. conceived, designed, and supervised the project and wrote the manuscript.

## Additional information

**Competing interests:** The authors declare no competing interests.

