## [Peer Review File · Nature Communications]

Reviewers' Comments:

Reviewer #1:

Remarks to the Author:

In this paper, the authors demonstrate a novel feedback loop involving RANK signaling, miR-182, PKR, and IFNB. The authors use a comprehensive approach that involves myeloid lineage-restricted miR-182 loss of function and gain of function mouse models. They demonstrate the ability of miR-182 levels to modify trabecular bone volume and osteoclast parameters in normal mice and in mice subjected to an ovariectomy (OVX) remodeling challenge. The ability of miR-182 loss of function to ameliorate bone loss in an inflammatory arthritis model was also demonstrated. With respect to translational significance, the authors demonstrate that administration of a miR-182 inhibitor using chitosan nanoparticles can prevent OVX-induced bone loss and bone loss in the inflammatory arthritis model.

Although PKR was suggested to be a miR-182 target in previous studies, the applicants make the connection between miR-182, PKR, and IFNB in osteoclast lineage cells. They demonstrate, using PKR-null mice, that PKR is a negative regulator of osteoclastogenesis, its effects mediated at least in part by IFNB activity. Lastly, the authors demonstrate correlations between miR-182 levels and anti-TNF therapy in a small cohort of rheumatoid arthritis patients.

Overall, this is a very nice study. Concerns are primarily related to what the authors have not shown, especially with regard to the in vivo mouse studies. These concerns are detailed below, and should be easily addressable.

1. Since there is cross talk between osteoblasts and osteoclasts, the authors should show data on bone formation rate and osteoblast parameters, determined by histomorphometry.
2. It is also important to show data on cortical bone parameters, not just the trabecular compartment.
3. The authors state that administration of miR-182 oligo inhibitor "completely reversed OVX induced bone loss". However, given the experimental design, in which the inhibitor was administered only 3 days after OVX surgery, it is more appropriate to state that the miRNA inhibitor prevented OVX induced bone loss.
4. miR-182 is negative regulator of osteoblast proliferation and differentiation. Because of this, it is important for the authors to also show bone formation rate and histomorphometric analysis of osteoblast parameters for the OVX studies shown in Figure 3. This can allow an estimation of the contribution of osteoblast vs osteoclast response to the miRNA inhibitor.
5. For the RNAseq studies, the authors should state how long the BMMs were treated with RANKL, and they should state the number of biological replicates examined.
6. Although PKR was pulled down in a biotinylated miR-182 IP performed by another group, the authors should perform 3'UTR luciferase reporter assays and mutational analysis to validate that PKR is directly targeted by miR-182.
7. The Western blot data shown in Figure 5c should be quantified and analyzed for statistical significance.

Smaller items:

1. The authors should state gender of mice used for in vitro BMM studies
2. In Supplemental Figure 1, the authors should show the degree to which miR-182 over expression was achieved in vitro and in vivo.
3. The image shown in Figure 2n is too small to be informative.
4. In Figures 5a, 5h and Supplemental Figure 2, it is more appropriate to show the osteoclast count data as the actual number of osteoclasts per well, rather than expressing the data as percent of control.
5. Are the calvarial osteolysis and serum transfer arthritis experiments also performed in female mice? The authors should state explicitly in the methods the gender of the mice used.

Reviewer #2:

Remarks to the Author:

Overall the manuscript has potential but appears to be a resubmission given that some experiments flow well in logical reasoning and support the primary hypotheses, while others appear tangential and unrelated to the data (e.g. Figure 6). I would recommend major revision and reconsideration.

The strengths of the manuscript are the demonstration of miR-182-modulation in osteoclastogenesis, and the documentation that its inhibition leads to a reduction in osteoclast formation and relevant gene expression in both mice and human cells. The authors also show a very nice RNAseq analysis leading them to focus on PKR and interferons. There are however a few critical avenues of investigation that would need to be completed prior to publication. The role of osteoblasts in the observed increases in bone mass with inhibition miR-182 remain unexplored. The scientific premise of a miR-182/PKR/IFN-beta loop would be significantly strengthened by experiments examining IFN-beta in miR-182 KO and Tg mice. In essence, scientifically the data to show loss of miR-182 modulation in IFN-beta KO mice would help definitively establish the cytokine as the critical downstream element.

Reviewer #3:

Remarks to the Author:

In this paper, Inoue et al. study miR-182 in bone destructive diseases. They identify this miRNA as a key regulator in osteoclastogenesis, employing multiple approaches in vitro and in mouse models in vivo. On the mechanistic level, PKR is identified as direct target and novel inhibitor in osteoclastogenesis that is instrumental in miR-182 action. The antimiR-mediated inhibition of miR-182, in vivo by using chitosan nanoparticles, protects mice from excessive osteoclastogenesis and suppresses pathologic bone erosion.

This is a comprehensive study on the mechanism of action of (aberrantly overexpressed) miR-182 and, on the preclinical stage, on therapeutic strategies based on the application of antimiRs that are formulated in nanoparticles.

Comments:

1- While the presented effects are well appreciated, some figures show differences in their magnitude. Compare +/- RANKL (white bars in Fig. 1c vs. Suppl. Fig. 1b): for example, Nfatc1 upregulation is almost 4-fold in Fig. 1c, but just 2-fold in Suppl. Fig. 1b; Prdm1 alterations are 5-fold vs. 10-fold and so on. Please explain these discrepancies.

2- In the Mir182(mTg) mouse, the amount of miR-182 overexpression should be stated.

3- Likewise, while it is positively noted that the authors apply rather small antimiR amounts (5 ug) and observe therapeutic effects after systemic (i.v.) application, it would be important to know how much antimiR is actually found in the target tissue / target cells. This reviewer agrees that a comprehensive biodistribution study would be beyond the scope of this paper, but the local antimiR concentration at its site of action is an important information that should be given. This is especially so since it seems that rather low levels may already be sufficient.

4- Following up on comment 1: When comparing Fig. 1h with Fig. 2c (sham), numbers are substantially different. Why? The same is true for Fig. 3c (control; sham vs. OVX) compared to Fig. 2c (WT; sham vs. OVX).

5- The authors state that they identify PKR as a direct target of miR-182. However, just to

“confirm the seed region of miR-182 in the 3'-UTR ... (Fig. 4f)” is not sufficient as a proof for ****direct**** targeting. The gold standard is a 3'-UTR reporter gene experiment with the direct comparison of the wt and a mutated miR-182 seed region in the PKR 3'UTR: only if miR-182 inhibitory effects on reporter gene expression are lost when mutating the seed region, PKR is proven as a direct target. Since in silico analyses, unfortunately, have often been found as not sufficiently predictive, this experiment will have to be done to establish PKR as a direct target.

6- In Fig. 6a, it is interesting to see that not all patients show miR-182 upregulation vs. healthy. Please comment. Does this indicate molecular differences in pathology between patients which may even serve as a predictor for therapy response? Are clinical differences in the patients seen that could be attributed or correlated to their different miR-182 levels?

7- The authors state that the miR-182 inhibitor is LNA modified. However, some background information (sequence with position(s) of the LNA modification(s)) would be helpful. The same is true for the negative control. Also, the volume used of i.v. injection of the nanoparticles should be given.

Response to Reviewers

NCOMMS-18-01258A: "Bone protection by inhibition of microRNA-182" by Kazuki Inoue, Zhonghao Deng, Yufan Chen, Eugenia Giannopoulou, Ren Xu, Shiaoqing Gong, Matthew B. Greenblatt, David G. Kirsch, Lingegowda S. Mangala, Gabriel Lopez-Berestein, Anil K. Sood, Liang Zhao and Baohong Zhao.

We thank the reviewers for their time and their positive and insightful comments. We are pleased that the reviewers were very enthusiastic. Reviewer #1: "a very nice study"; "a novel feedback loop involving RANK signaling, miR-182, PKR, and IFNB"; "a comprehensive approach"; "translational significance". Reviewer #2: "very nice RNAseq analysis"; "overall interesting clinically". Reviewer #3: "a comprehensive study on the mechanism of action of miR-182 and, on the preclinical stage, on therapeutic strategies based on the application of anti-miRs that are formulated in nanoparticles". We have experimentally addressed the points raised by the reviewers and generated 39 new figure panels of data. We have revised the manuscript accordingly and the reviewers' points are specifically addressed below. Changes in the manuscript have been underlined.

Response to specific points:

Reviewer #1 (Remarks to the Author):

In this paper, the authors demonstrate a novel feedback loop involving RANK signaling, miR-182, PKR, and IFNB. The authors use a comprehensive approach that involves myeloid lineage-restricted miR-182 loss of function and gain of function mouse models. They demonstrate the ability of miR-182 levels to modify trabecular bone volume and osteoclast parameters in normal mice and in mice subjected to an ovariectomy (OVX) remodeling challenge. The ability of miR-182 loss of function to ameliorate bone loss in an inflammatory arthritis model was also demonstrated. With respect to translational significance, the authors demonstrate that administration of a miR-182 inhibitor using chitosan nanoparticles can prevent OVX-induced bone loss and bone loss in the inflammatory arthritis model. Although PKR was suggested to be a miR-182 target in previous studies, the applicants make the connection between miR-182, PKR, and IFNB in osteoclast lineage cells. They demonstrate, using PKR-null mice, that PKR is a negative regulator of osteoclastogenesis, its effects mediated at least in part by IFNB activity. Lastly, the authors demonstrate correlations between miR-182 levels and anti-TNF therapy in a small cohort of rheumatoid arthritis patients. Overall, this is a very nice study. Concerns are primarily related to what the authors have not shown, especially with regard to the in vivo mouse studies. These concerns are detailed below, and should be easily addressable.

We thank the reviewer for his/her very positive comments and for bringing out these important points. We have performed experiments and addressed the reviewer's questions as below.

Questions 1 and 4: *Since there is cross talk between osteoblasts and osteoclasts, the authors*

should show data on bone formation rate and osteoblast parameters, determined by histomorphometry. Q4: miR-182 is negative regulator of osteoblast proliferation and differentiation. Because of this, it is important for the authors to also show bone formation rate and histomorphometric analysis of osteoblast parameters for the OVX studies shown in Figure 3. This can allow an estimation of the contribution of osteoblast vs osteoclast response to the miRNA inhibitor.

Following the reviewer's suggestion, we have performed calcein double labeling of newly formed bones in mice and further performed dynamic and static histomorphometric analyses of the mouse bone specimen to measure bone formation rate and osteoblast parameters. We found that osteoclastic deletion of miR-182 (*Mir182^{ΔM/ΔM}* mice vs. the control mice) did not significantly affect bone formation rate or osteoblast parameters, such as osteoblast numbers and osteoblast surfaces (Supplementary Fig. 3b, c). Furthermore, osteoclastic miR-182 deficiency (*Mir182^{ΔM/ΔM}* mice vs. the control mice, Supplementary Fig. 5b, c) or treatment with the CH-nanoparticles containing miR-182 inhibitors (Supplementary Fig. 9b, c) did not affect bone formation rate, osteoblast numbers and osteoblast surfaces in the sham or OVX mice during the experimental period. These data suggest that specific inhibition of miR-182 in myeloid lineage does not have significant impact on bone formation rate and osteoblast parameters. These new results are now shown in Supplementary Fig. 3b, c, Supplementary Fig. 5b, c and Supplementary Fig. 9b, c, and noted in the text on pp. 7, 8, 10 and 21.

2. It is also important to show data on cortical bone parameters, not just the trabecular compartment.

As suggested by the reviewer, we performed microCT analysis of the midshaft of femurs to determine the cortical bone phenotype. We found that the cortical bone thickness was not altered by osteoclastic miR-182 deficiency (Supplementary Fig. 3a). Furthermore, osteoclastic miR-182 deficiency (*Mir182^{ΔM/ΔM}* mice vs. the control mice, Supplementary Fig. 5a) or treatment with the CH-nanoparticles containing miR-182 inhibitors (Supplementary Fig. 9a) did not affect cortical thickness in the sham or OVX mice during the experimental period. These findings indicate that inhibition of miR-182 in the myeloid lineage does not affect cortical compartment. These new results are now shown in Supplementary Fig. 3a, Supplementary Fig. 5a and Supplementary Fig. 9a, and noted in the text on pp. 7, 8, 10 and 21.

3. The authors state that administration of miR-182 oligo inhibitor "completely reversed OVX induced bone loss". However, given the experimental design, in which the inhibitor was administered only 3 days after OVX surgery, it is more appropriate to state that the miRNA inhibitor prevented OVX induced bone loss.

We thank the reviewer for suggesting a more appropriate wording and have accordingly changed text to "prevented" on pg. 10.

5. For the RNAseq studies, the authors should state how long the BMMs were treated with RANKL, and they should state the number of biological replicates examined.

Following the reviewer's suggestion, the RANKL treatment time and the detailed information for the RNAseq conditions and samples, which were obtained from the mRNAs extracted from the

WT and *Mir182*^{AM/AM} BMMs, or the Control and *Mir182*^{mTg} BMMs (complementary cell cultures), as well as the number of biological replicates are now stated in the Fig. 4 legend on pg. 37.

6. *Although PKR was pulled down in a biotinylated miR-182 IP performed by another group, the authors should perform 3'UTR luciferase reporter assays and mutational analysis to validate that PKR is directly targeted by miR-182.*

We thank the reviewer for this constructive suggestion. Following this line, we performed a 3' UTR luciferase reporter assay, and found that miR-182 mimic down-regulated the luciferase activity of the 3' UTR of *Eif2ak2* (PKR) reporter (Fig. 4i). Mutation of the seed region of miR-182 in the 3' UTR of *Eif2ak2* (PKR) abolished the regulatory effect of miR-182 on the 3' UTR of *Eif2ak2* (PKR) (Fig. 4i). These results collectively validate that PKR is directly targeted by miR-182. These results are now shown in Fig. 4i, and noted on pp. 12 and 25.

7. *The Western blot data shown in Figure 5c should be quantified and analyzed for statistical significance.*

The relative density of the immunoblot bands of NFATc1, Blimp1 and PKR vs. those of loading control p38 from three independent experiments were quantified by densitometry and the statistical analysis was performed. The data is now shown in supplementary Fig. 11, and noted on pp. 12 and 44.

Smaller items:

1. *The authors should state gender of mice used for in vitro BMM studies.*

We used age and gender-matched mutant and control mice within each experiment throughout in vitro BMM studies. Following the reviewer's suggestion, this point has been noted in the *Cell culture* section in *Methods* on pg. 22.

2. *In Supplemental Figure 1, the authors should show the degree to which miR-182 over expression was achieved in vitro and in vivo.*

The relative expression levels of miR-182 in the bone marrow macrophages derived from the bone marrow (in vitro) or the osteoclast precursors (in vivo) isolated from the *Mir182*^{mTg} and the control mice were examined and the overexpression of miR-182 in *Mir182*^{mTg} was confirmed. The data is now shown in the new Supplementary Fig. 2a (corresponding to the old Supplementary Figure 1).

3. *The image shown in Figure 2n is too small to be informative.*

The image in Figure 2n is now enlarged.

4. *In Figures 5a, 5h and Supplemental Figure 2, it is more appropriate to show the osteoclast count data as the actual number of osteoclasts per well, rather than expressing the data as percent of control.*

Following the reviewer's suggestion, the osteoclast count data are shown in Figures 5a, 5h, and new Supplementary Fig. 6 (previous Supplementary Fig. 2).

5. Are the calvarial osteolysis and serum transfer arthritis experiments also performed in female mice? The authors should state explicitly in the methods the gender of the mice used.

The female mice were used in the serum transfer arthritis models, and the age and gender-matched mutant and their control mice were used in the calvarial osteolysis experiments. These points are now added in the *Methods* section on pg. 20.

Reviewer #2 (Remarks to the Author):

Reviewer 2 comments:

The main premise is that during osteoclast development, RANKL induces the expression of miR-182, which in turn inhibits PKR. PKR appears to positive regulate IFN-beta production in a variety of settings, and for osteoclastogenesis, PKR-induced IFN-beta acts as a negative autocrine regulator.

The authors demonstrate miR-182 expression in osteoclasts and that its inhibition leads to a reduction in osteoclast formation and relevant gene expression in both mice and human cells. Moreover, they show that ovariectomy-induced bone loss is not manifest in mice where miR-182 is inhibited. They go on to show that, in mice overexpressing miR-182, there is excessive osteoclastogenesis in TNF-related models of bone destruction. The characterization of miR-182 transgenic animals is placed as supplemental figure 1, but is in line with expectations of miR-182 being pro-osteoclastogenic. The authors further show very nice RNAseq analysis using wildtype, miR-182 inhibited, and miR-182 Tg animals to zero in on pathways and genes affected by miR-182. From this they focus on PKR and interferons. They attempt to show that miR-182 inhibits PKR, which is needed for IFN-beta expression (a negative regulator of osteoclast formation). Figure 6 extends this correlation in RA patients, and is overall interesting clinically; however, the focus of Figure 6 is slightly different from the main story in that Figure 6 is more focused to the role of miR-182 in TNF-mediated inflammation and how this correlates with IFN-beta rather than osteoclastogenesis. Despite an overall rigorous evaluation, there are some issues needing consideration.

We thank the reviewer for his/her insightful summary and enthusiastic comments on our work. We followed and experimentally addressed the reviewer's constructive questions below and believe these new data significantly improved and strengthened our manuscript.

1. The increases in bone mass with inhibition miR-182 expression need osteoblast/bone formation analysis (e.g. what are the bone formation measures on histomorphometry). There is no data shown in either the main figures or the supplemental ones in regards to if/how osteoblasts are affected.

Following the reviewer's suggestion, we have performed calcein double labeling of newly

formed bones in mice and further performed dynamic and static histomorphometric analyses of the mouse bone specimen to measure bone formation rate and osteoblast parameters. We found that osteoclastic deletion of miR-182 (*Mir182^{ΔM/ΔM}* mice vs. the control mice) did not significantly affect bone formation rate or osteoblast parameters, such as osteoblast numbers and osteoblast surfaces (Supplementary Fig. 3b, c), indicating that the specific inhibition of miR-182 in myeloid lineage does not have significant impact on bone formation rate and osteoblast parameters. These new results are now shown in Supplementary Fig. 3b, c, and noted in the text on pp. 7 and 21.

2. The authors should look at the ability of increasing concentrations of osteoclastic cytokines to overcome the loss of miR-182. They need to use cells from the KO, WT and Tg mice and do a RANKL concentration curve for osteoclast formation. If, as the authors suggest, miR-182 is a feedforward agent for RANKL-induced osteoclastogenesis, can they see “normal” levels of formation at higher RANKL concentrations in the KO, and likewise “normal” levels of formation in the Tg at lower RANKL concentrations?

Following the reviewer's suggestion, we used BMMs derived from WT and *Mir182^{ΔM/ΔM}* mice (Supplementary Fig. 1), or from the Control and *Mir182^{mTg}* mice (Supplementary Fig. 2d) to examine osteoclast differentiation induced by RANKL at a serial of different concentrations. We found that inhibition of osteoclastogenesis by miR-182 deficiency was observed in the presence of RANKL at different concentrations (Supplementary Fig. 1). High concentrations of RANKL are required to induce osteoclastogenesis in the miR-182 deficient cells, for example, RANKL at 120 ng/ml can induce osteoclast differentiation in the miR-182 KO cell cultures to a similar level as that induced by 40 ng/ml of RANKL in the WT cell cultures (“normal” level of differentiation) (Supplementary Fig. 1). On the other hand, overexpression of miR-182 enables RANKL to induce efficient osteoclast differentiation at lower concentrations with approximately 25-50% of those used in the control cell cultures (Supplementary Fig. 2d). These results support the finding that miR-182 functions as a RANKL-inducible feed forward regulator to promote osteoclast differentiation. These new results are now shown in Supplementary Fig. 1 and Supplementary Fig. 2d, and discussed on pp. 6 and 17.

Question 2 continued: An alternative to concentration curve studies would be to conduct the IFN-antibody inhibition experiments in the miR-182 KO and Tg animals; if you neutralize the IFN-beta in the KO animals, is there a re-emergence of osteoclastogenesis? Similarly, does neutralization of IFN-beta have no additional effect in Tg animals?

To address these questions, we followed the reviewer's suggestion to conduct the IFN-antibody inhibition experiments in the miR-182 KO and Tg cell cultures. We found that blocking of endogenous IFN- β using an IFN- β neutralizing antibody (Fig. 6c) or by knocking down IFN- β expression (Fig. 6d, e) abrogated the inhibitory effect of miR-182 deficiency on the RANKL-induced osteoclastogenesis. On the other hand, neutralization of endogenous IFN- β did not have additional effect on the miR-182 overexpression-enhanced osteoclastogenesis in the *Mir182^{mTg}* cell cultures (Fig. 6f). These findings indicate that IFN- β is a critical downstream component of miR-182 during osteoclastogenesis. These new results are now shown in the Fig. 6c, d, e, f, and discussed on pg. 13.

3. There needs to be an explanation for differing data: in the basal state of the animal models,

miR-182 inhibition leads to an elevation in bone mass from osteoclastic suppression (e.g. in Figure 2, the sham animals without miR-182 have an elevated bone mass compared to wildtype controls). In Figure 3, they show that miR-182 inhibition via nanoparticles prevents OVX bone loss but interestingly does not appear to affect the sham-operated animals. Why, if miR-182 promotes normal osteoclastic development, is there no significant change in the sham-operated case with its inhibition?

In the OVX model, the period of treatment with nanoparticles containing miR-182 inhibitor is 5 weeks, meaning a short term inhibition of miR-182. In the sham groups, CH-nanoparticles containing miR-182 inhibitors did not affect basal bone mass during this short treatment period. It is however possible that long term treatment with miR-182 inhibitors may increase bone mass, as observed in the miR-182 deficient mice. It will be of interest to investigate the effect of a long term treatment with the miR-182 inhibitor on bone mass in future studies.

4. Would be great to show loss of miR-182 modulation in IFN-beta KO mice, definitively establishing that the critical downstream element is IFN-beta. At a minimum, I would recommend showing that IFN-beta levels are up-modulated in the miR-182 inhibited animals.

Following the reviewer's suggestion, we examined IFN- β expression and found that miR-182 deficiency significantly enhanced the RANKL-induced IFN- β levels (Fig. 6a). Importantly, the protein levels of IFN- β in the serum were also markedly increased in the *Mir182*^{ΔM/ΔM} mice (Fig. 6b). We furthermore tested the functional importance of the up-regulated IFN- β by miR-182 deficiency. We found that blocking of endogenous IFN- β using an IFN- β neutralizing antibody (Fig. 6c) or by knocking down IFN- β expression (Fig. 6d, e) abrogated the inhibitory effect of miR-182 deficiency on the RANKL-induced osteoclastogenesis. These findings collectively indicate that IFN- β is a critical downstream element of miR-182 during osteoclastogenesis. These new results are now shown in the Fig. 6a, b, c, d and e, and discussed on pg. 13.

5. Supplementary Figure 2 needs comparison to the unprimed conditions in order to be able to interpret if priming is adjusting the response.

We added unprimed conditions in Supplementary Fig. 6 (previous Supplementary Fig. 2), and found that miR-182 deficiency significantly inhibited osteoclast differentiation in both RANKL alone condition and RANKL priming condition (Middle panel, Supplementary Fig. 6). It is well known that TNF alone cannot efficiently induce osteoclast differentiation, which was also observed and confirmed in our experiments (Right panel, Supplementary Fig. 6). Thus, it is not an appropriate approach using TNF alone stimulation to examine inhibition of osteoclastogenesis. Therefore, we used TNF priming followed by TNF and RANKL costimulation, a common culture method to mimic an inflammatory setting. As shown in the right panel in Supplementary Fig. 6, miR-182 deficiency significantly inhibited osteoclast differentiation in the TNF priming condition followed by TNF and RANKL costimulation. Thus, miR-182 deletion plays a similar inhibitory role in these priming conditions as in RANKL unprimed condition. These results are now shown in Supplementary Fig. 6, and noted on pg. 43.

Reviewer #3 (Remarks to the Author):

In this paper, Inoue et al. study miR-182 in bone destructive diseases. They identify this miRNA as a key regulator in osteoclastogenesis, employing multiple approaches in vitro and in mouse models in vivo. On the mechanistic level, PKR is identified as direct target and novel inhibitor in osteoclastogenesis that is instrumental in miR-182 action. The antimiR-mediated inhibition of miR-182, in vivo by using chitosan nanoparticles, protects mice from excessive osteoclastogenesis and suppresses pathologic bone erosion. This is a comprehensive study on the mechanism of action of (aberrantly overexpressed) miR-182 and, on the preclinical stage, on therapeutic strategies based on the application of antimiRs that are formulated in nanoparticles.

We thank the reviewer for his/her insightful summary and positive comments on our study. We have experimentally addressed the reviewer's constructive questions below as well as explained and clarified some important points the reviewer brought out. We believe these new data and discussion significantly improved and strengthened our manuscript.

Comments:

1- While the presented effects are well appreciated, some figures show differences in their magnitude. Compare +/- RANKL (white bars in Fig. 1c vs. Suppl. Fig. 1b): for example, Nfatc1 upregulation is almost 4-fold in Fig. 1c, but just 2-fold in Suppl. Fig. 1b; Prdm1 alterations are 5-fold vs. 10-fold and so on. Please explain these discrepancies.

The discrepancy in the relative expression magnitude of these osteoclast genes is related to the differences in the culture times with RANKL (kinetic difference in gene expression) in different experiments according to different experimental purposes (for example, prolonging culture times to see inhibition of osteoclast differentiation). The culture time in Fig. 1c is 5 days, and in the new Supplementary Fig. 2c (previous Supplementary Fig. 1b) is 2 days. Within each experiment, since we cultured the cells on the same plates for the same time periods together with corresponding controls, we believe the comparisons of gene expression between conditions are valid. In addition, the induction patterns of those genes in the control cultures show similar trends between experiments. To clarify this point and show data more informative, we have indicated the culture times in each relevant figure legend on pp. 35 and 42.

2- In the Mir182(mTg) mouse, the amount of miR-182 overexpression should be stated.

The relative expression levels of miR-182 in the bone marrow macrophages derived from the bone marrow (in vitro) or the osteoclast precursors (in vivo) isolated from the *Mir182^{mTg}* and the control mice were examined and the overexpression of miR-182 in *Mir182^{mTg}* was confirmed. The data is now shown in the new Supplementary Fig. 2a (corresponding to the old Supplemental Figure 1).

3- Likewise, while it is positively noted that the authors apply rather small antimiR amounts (5 ug) and observe therapeutic effects after systemic (i.v.) application, it would be important to know how much antimiR is actually found in the target tissue / target cells. This reviewer agrees that a comprehensive biodistribution study would be beyond the scope of this paper, but the local

antimiR concentration at its site of action is an important information that should be given. This is especially so since it seems that rather low levels may already be sufficient.

We agree with the reviewer that it is important to know if small antimiR amounts (5ug) have sufficient effect on lowering the levels of miR-182 in the target tissue bone marrow. To address this question, we have tested and compared the expression levels of miR-182 in the bone marrow isolated from the mice treated with CH nanoparticles containing the control or miR-182 inhibitor (antimiR-182), and found that the local miR-182 expression level in bone marrow was decreased approximately 70% by the small amount of anti-miR-182. This piece of important data is now shown in Supplementary Fig. 8e and noted on pg. 18.

4- Following up on comment 1: When comparing Fig. 1h with Fig. 2c (sham), numbers are substantially different. Why? The same is true for Fig. 3c (control; sham vs. OVX) compared to Fig. 2c (WT; sham vs. OVX).

The values of bone mass and bone parameters, such as trabecular bone volume, bone mineral density, connectivity density, trabecular bone number and trabecular bone spacing, are closely related to mouse gender, age, strains, genetic background as well as environments (breeding environments and nutrition etc). The mouse gender and age in Fig. 1h and Fig. 2c are different. In Fig. 1h, the parameters were obtained from 10 week old **male** mice. The OVX model in Fig. 2c required usage of female mice. The bone phenotype of 15 week old **female** mice was analyzed for the sham or OVX groups in Fig. 2c. In Fig. 3c and Fig. 2c, the mouse gender and age are the same because the OVX model was used. However, the strain background in Fig. 3c is WT C57BL6. In Fig. 2c, the "WT" indicates the control mice, which has a *Mir182*^{+/+}*LysMcre*(+) genotype (this point has been noted in *Results* section on pg. 6 and the *Methods* section on pg. 20). According to different experimental purposes, the mice with different genetic background were used. For example, to investigate the role of miR-182, we applied both genetic ablation approach (Fig. 2c) and pharmacological inhibitors (Fig. 3c). Accordingly, in Fig. 2c, the *Mir182*^{flox/flox}*LysMcre*(+) mice (referred to as *Mir182*^{ΔM/ΔM}) and the control mice with a *Mir182*^{+/+}*LysMcre*(+) genotype (referred to as WT) were used. In Fig. 3c, C57BL6 mice were used. Because of these different factors, the bone parameter values show variation in different experiments. Within each experiment, we have controlled mouse gender, age, strain, genetic background and environment to be same. Therefore, the comparisons of the bone parameters between conditions in each experiment are valid. These important points related with mouse gender, age and genetic background are now explicitly noted in each relevant figure legend as well as in related *Results* parts and in *Methods* section.

*5- The authors state that they identify PKR as a direct target of miR-182. However, just to "confirm the seed region of miR-182 in the 3'-UTR ... (Fig. 4f)" is not sufficient as a proof for **direct** targeting. The gold standard is a 3'-UTR reporter gene experiment with the direct comparison of the wt and a mutated miR-182 seed region in the PKR 3'UTR: only if miR-182 inhibitory effects on reporter gene expression are lost when mutating the seed region, PKR is proven as a direct target. Since in silico analyses, unfortunately, have often been found as not sufficiently predictive, this experiment will have to be done to establish PKR as a direct target.*

We thank the reviewer for this constructive suggestion. Following this line, we performed a 3' UTR luciferase reporter assay, and found that miR-182 mimic down-regulated the luciferase

activity of the 3' UTR of *Eif2ak2* (PKR) reporter (Fig. 4i). Mutation of the seed region of miR-182 in the 3' UTR of *Eif2ak2* (PKR) abolished the regulatory effect of miR-182 on the 3' UTR of *Eif2ak2* (PKR) (Fig. 4i). These results collectively validate that PKR is directly targeted by miR-182. These results are now shown in Fig. 4i, and noted on pp. 12 and 25.

[Redacted]

7- The authors state that the miR-182 inhibitor is LNA modified. However, some background information (sequence with position(s) of the LNA modification(s)) would be helpful. The same is true for the negative control. Also, the volume used of i.v. injection of the nanoparticles should be given.

Following the reviewer's suggestion, some background information regarding the LNA modification and sequences has been added in the Methods section on pg. 22. The volume (100 ul) used of i.v. injection of the nanoparticles is now noted in the Methods section on pg. 21.

Response to Reviewers

NCOMMS-18-01258A: "Bone protection by inhibition of microRNA-182" by Kazuki Inoue, Zhonghao Deng, Yufan Chen, Eugenia Giannopoulou, Ren Xu, Shiaoqing Gong, Matthew B. Greenblatt, Lingegowda S. Mangala, Gabriel Lopez-Berestein, David G. Kirsch, Anil K. Sood, Liang Zhao and Baohong Zhao.

We thank the reviewers for their time and their very positive comments on our revised manuscript. We are pleased that Reviewer 2 and 3 are overall satisfied with the revised manuscript, and Reviewer 1 "*feel that the paper is suitable for publication*" with only a minor suggestion for additional RNAseq replicates. Following Reviewer 1's suggestion and the editor's comment, we have performed additional RNAseq experiments and provided biological replicates for each condition in all RNAseq experiments. The reproducibility between the biological RNAseq replicates for each condition is significantly high with Pearson's $R \geq 0.986$. The conclusions are not changed with the additional biological RNAseq replicates. We have included the replicates and made minor changes in the text. The reviewers' points are specifically addressed below. Changes in the text have been highlighted in yellow color.

Response to specific points:

Reviewer #1 (Remarks to the Author):

The authors have been very responsive to the concerns of the reviewers, providing the additional data and descriptions of the experiments, as requested. With these additional descriptions of the experiments, it is now clear that the RNAseq data shown represent N=1 for each condition. This is OK (but risky) for hypothesis generation, but it is unacceptable for publication. The authors should either repeat the sequencing studies with additional biological replicates or remove the data and their discussion from the manuscript. Other than this concern, I feel that the paper is suitable for publication.

We thank the reviewer for bringing out this important point. We have performed additional independent RNAseq experiments. We further performed Pearson correlation analysis to assess the reproducibility of RNAseq data. As shown in the new Supplementary Fig. 11, the gene expression values of the two independent biological RNAseq replicates are highly correlated for each condition with Pearson's $R \geq 0.986$, indicating a markedly high reproducibility between these replicates for each condition. We then analyzed the RNAseq data using these replicates. We are very happy that the conclusions are not changed with the additional biological RNAseq replicates. Furthermore, we found that the replicates even increased the significance, for example, the p values and FDR values with the replicates indicate that the results are more significant. The conclusions are not changed. We showed the reproducibility analysis data in supplementary Fig. 11 and labeled minor changes of the text according to the results with enhanced significance on pp. 11, 12, 25, 38, 45 and 46.

Reviewer #2 (Remarks to the Author): *The authors have thoughtfully and thoroughly responded to all critique. Please show low magnification fluorescence micrographs for calcein labeling studies, rather than artificially selected areas currently shown at a high magnification. Otherwise, there are no further issues.*

We are happy that the reviewer is satisfied with the revision. The low magnification fluorescence micrographs for calcein labeling have now been shown in Supplementary Fig. 3, 5 and 9.

Reviewer #3 (Remarks to the Author): *The authors have fully addressed this reviewer's issues and comments, and made appropriate additions to the figures and the text.*

We are happy that the reviewer is satisfied with the revision.

Response to Reviewers

NCOMMS-18-01258: "Bone protection by inhibition of microRNA-182" by Kazuki Inoue, Zhonghao Deng, Yufan Chen, Eugenia Giannopoulou, Ren Xu, Shiaoqing Gong, Matthew B. Greenblatt, Lingegowda S. Mangala, Gabriel Lopez-Berestein, David G. Kirsch, Anil K. Sood, Liang Zhao and Baohong Zhao.

Reviewer #1 (Remarks to the Author): The additional replicate is adequate.

We are happy that the reviewer is satisfied with the revision.